# GUARD: Constructing Realistic Two-Player Matrix and Security Games for Benchmarking Game-Theoretic Algorithms

**Noah Krever, Jakub Černý, Moïse Blanchard, Christian Kroer**
Department of Industrial Engineering and Operations Research
Columbia University
New York, NY 10027
{ndk2115,jakub.cerny,mb5414,christian.kroer}@columbia.edu

## Abstract

Game-theoretic algorithms are commonly benchmarked on recreational games, classical constructs from economic theory such as congestion and dispersion games, or entirely random game instances. While the past two decades have seen the rise of security games – grounded in real-world scenarios like patrolling and infrastructure protection – their practical evaluation has been hindered by limited access to the datasets used to generate them. In particular, although the structural components of these games (e.g., patrol paths derived from maps) can be replicated, the critical data defining target values – central to utility modeling – remain inaccessible. In this paper, we introduce a flexible framework that leverages open-access datasets to generate realistic matrix and security game instances. These include animal movement data for modeling anti-poaching scenarios and demographic and infrastructure data for infrastructure protection. Our framework allows users to customize utility functions and game parameters, while also offering a suite of preconfigured instances. We provide theoretical results highlighting the degeneracy and limitations of benchmarking on random games, and empirically compare our generated games against random baselines across a variety of standard algorithms for computing Nash and Stackelberg equilibria, including linear programming, incremental strategy generation, and self-play with no-regret learners.

## 1 Introduction

Equilibrium finding in games has become a cornerstone of artificial intelligence, powering breakthroughs in recreational play (e.g., poker, Stratego, Diplomacy) [5, 8, 38, 2] as well as critical applications in security and logistics [39, 20, 44, 25, 11]. At the heart of these successes are efficient algorithms for computing game-theoretic equilibria. To develop, evaluate, and compare such methods, practitioners typically turn to benchmarks drawn from recreational games, classical economic models, or pseudo-random payoff matrices. Yet among these standard baselines, the model that has seen the greatest real-world success, the Stackelberg security game model, is conspicuously absent.

Security games, where a defender allocates limited resources across targets and an attacker selects among them, have guided deployment of patrols across the eight terminals of Los Angeles International Airport [39], helped protect biodiversity across 2,500 km$^2$ of conservation area [20, 18, 19] and screening more than 800 million US air passengers annually [7]. Extensions have been developed for traffic monitoring, drug interdiction, and cybersecurity [43]. Practical evaluation of game-theoretic algorithms on security games, however, has been hindered by limited access to the most valuable component: target-value utilities. Although the structural components–map-based patrol routes and

scheduling constraints–can be reconstructed, till this day, the utilities that encode real-world strategic trade-offs remain largely inaccessible to algorithm developers.

To fill this gap in benchmarking ability, we introduce GUARD,[1] a flexible framework that constructs realistic matrix and security game instances from open-access data. Drawing on readily available sources such as animal movement records for anti-poaching scenarios and demographic or infrastructure data for critical asset protection, our framework supports both custom game specification and access to a library of ready-made instances, enabling meaningful and reproducible evaluation of game-theoretic algorithms on security-inspired scenarios. The framework provides a more realistic real-world resource allocation and security grounded alternative to existing game generators such as Gamut [36], which includes more traditional and classical economic games, or OpenSpiel [30], which offers a wide range of recreational and synthetic benchmarks. A more detailed explanation of the differentiation between Gamut and GUARD is provided in C.9. Our game instances can be exported in formats compatible with tools like OpenSpiel and Gambit [40], and can also provide target values for integration into pursuit-evasion frameworks such as GraphChase [50].

In addition, we examine the implications of using randomly generated games as benchmarks, a common approach in the literature. In Section 3, we show that Stackelberg equilibria in random games exhibit very sparse or degenerate solutions, which are not expected in realistic games. Specifically, we show that for random utilities drawn from the uniform distribution, near-optimal utility for the defender can already be achieved with pure (or very sparse) strategies for random normal-form games, and with very few resources for random security games. Notably, this suggests that random Stackelberg games typically give unreasonable advantage to the defender, effectively ignoring the impact of the adversary. In Section 5, we complement these findings with empirical results, comparing our realistic game instances with random ones. We observe similar degeneracy not only in random general-sum games, but also in zero-sum settings and even in games played on realistic maps, where randomness is limited to the assignment of target values.

## 2 Game Representations and Solution Concepts

**Normal-form games.** We study two-player games in which Player 1 has $n$ actions and Player 2 has $m$ actions. In the normal-form (or matrix) representation, each player's payoffs are encoded by matrices $A, B \in \mathbb{R}^{n \times m}$. Entry $A_{ij}$ (respectively $B_{ij}$) gives the payoff to Player 1 (resp. Player 2) when the players choose actions $i$ and $j$. Each player may randomize over their actions; their mixed strategy spaces are the simplexes $\Delta^n$ and $\Delta^m$, where $\Delta^n = \{x \in \mathbb{R}_+^n \mid \sum_{i=1}^n x_i = 1\}$. For any strategy $x \in \mathbb{R}^d$, we define its support as $\mathrm{supp}(x) = \{i \in [d] \mid x_i \neq 0\}$. The *best responses* of Player 2 to a mixed strategy $x \in \Delta^n$ is the set $\mathrm{BR}_2(x) = \mathrm{argmax}_{y \in [m]} x^T B y$; ties are resolved arbitrarily. Player 1's best responses to strategies of Player 2 are denoted analogously as $\mathrm{BR}_1(y)$.

**Security games.** Beyond matrix games, we also consider security games, which model defender-attacker interactions over a set of targets $T$. The *defender* (Player 1) controls a set of resources $R$, each of which can be assigned to a schedule from a set $S \subseteq 2^T$, where a schedule specifies a subset of targets simultaneously protected by a single resource. A target is deemed "covered" if some assigned schedule includes it. The *attacker* (Player 2) selects targets to attack. In the standard formulation, the players' utilities depend only on whether the attacked target is covered. The defender receives a utility of $u_d^c(t)$ if a covered target $t$ is attacked, and $u_d^u(t)$ if it is not covered. The attacker similarly obtains $u_a^c(t)$ or $u_a^u(t)$ depending on coverage. If multiple attacks are allowed, utilities are aggregated equally across targets. When the resources in $R$ are not identical, a mapping $A : R \to S$ may define the valid schedules for each resource. We refer to the tuple $(A, u^u, u^c)$ as the *schedule form* of the game. It can be expanded into a normal-form game by enumerating all (exponentially many) actions.

**Nash equilibrium (NE) in zero-sum games.** In a zero-sum game, $A = -B$. Finding a Nash equilibrium $(x^*, y^*)$ in a two-player zero-sum game can be formulated as a saddle-point problem using the first player's payoff matrix. In particular, von Neumann's *minimax theorem* shows that $(x^*, y^*)$ is achieved by a saddle point of $x^T A y$,

$$\max_{x \in \Delta^n} \min_{y \in \Delta^m} x^T A y = \min_{y \in \Delta^m} \max_{x \in \Delta^n} x^T A y = x^{*T} A y^*.$$

---

[1]Code open-sourced and publicly-available at `https://github.com/CoffeeAndConvexity/GUARD`.

The minimax theorem shows that $(x^*, y^*)$ is a NE if and only if each mixed strategy optimizes the players' payoff assuming that the other player best responds. Two-player zero-sum games can be solved efficiently using a variety of methods, including fictitious play [6], linear programming [42], strategy generation [33], or self-play with no-regret learners [24].

**Strong Stackelberg equilibrium (SSE) in general-sum games.** In general-sum two-player games, where $A$ and $B$ may be arbitrary $n \times m$ matrices, a Stackelberg equilibrium $(x^*, y^*)$ can be formulated as a bilevel problem [48, 13]:

$$x^* = \operatorname*{argmax}_{x \in \Delta^n} x^T A y^*(x), \text{ where } \quad y^*(x) \in \mathrm{BR}_2(x) = \operatorname*{argmax}_{y \in [m]} x^T B y.$$

The equilibrium is considered *strong* if, in addition, Player 2 breaks ties in favor of Player 1. In zero-sum games, NE and SSE coincide for Player 1, i.e., the optimal SSE strategy for Player 1 is also a NE strategy. Beyond zero-sum, SSE can be computed efficiently using a sequence of linear programs in both two-player normal-form games [13] and security games with single-target schedules [29].

**Sparsity in game theoretic equilibria.** In both normal-form and security games, the support of a strategy reflects the strategic complexity of the equilibrium. Sparse strategies (with small supports) can indicate focused and interpretable behavior, and are often desirable for execution in real-world settings where certain constraints may limit the feasibility of complex strategies. However, extreme sparsity may reflect an underlying degeneracy in the game structure, particularly in synthetic benchmarks where unrealistic structures may result in equilibria that incorrectly simplify the strategy space. Conversely, overly dense strategies may offer only marginal gains from added complexity, leading to diminishing returns, potential redundancy, and impractical deployment in real systems. In this sense, the support sizes of equilibrium strategies serve as a diagnostic for evaluating whether a game meaningfully captures strategic nuance, thus becoming a useful empirical metric in the assessment of benchmark game realism.

## 3 Limitations of Benchmarking on Random Games

Random games often admit very sparse equilibria, making them tractable in ways that real-world instances are not. For example, while finding a Nash equilibrium in two-player general-sum games is PPAD-hard [12], random games with suitable payoff distributions can be solved in expected polynomial time [3]. More broadly, multi-player games with i.i.d. payoffs almost surely have a pure Nash equilibrium (approaching probability $1 - 1/e$ as action sets grow) [23, 16], and two-player Gaussian or uniform games admit equilibria supported on just two actions with probability $1 - O(1/\log n)$ [3]. In fact, the chance that a random two-player game has no equilibrium of support size $k$ decays exponentially in $k$, so brute-force support enumeration runs in expected polynomial time. We demonstrate that the same degeneracy arises in SSE too.

**Random general-sum normal-form games.** We assume that all elements in the utility matrices $A$ and $B$ are sampled i.i.d. uniformly in $[0, 1]$. For any Player 1 strategy $x \in \Delta^n$, let us denote by $V(x)$ any possible value Player 1 can obtain after Player 2 best-responds to $x$, i.e., $V(x) \in \{x^T A y, y \in \mathrm{BR}_2(x)\}$. Since all payoffs are uniform in $[0, 1]$, we always have $V(x) \in [0, 1]$. In particular, the SSE satisfies $V(x^*) \leq 1$. We analyze the performance of sparse strategies for Player 1. To this end, we define by $x^*(k)$ their best $k$-sparse Stackelberg equilibrium strategy, i. e., $\mathrm{supp}(x^*(k)) \leq k$, that maximizes the value $V$ under any fixed tie-breaking rule. The proof is given in Appendix A.1.

**Theorem 1.** *Let $A, B$ be sampled i.i.d. uniformly in $[0, 1]$. Then (i) $V(x^*(1)) \sim Beta(n, 1)$, and for every $C \geq 0$, $\mathbb{E}_{A,B}[V(x^*(1))] = 1 - \frac{1}{n+1}$ and $\mathbb{P}_{A,B}\left[V(x^*(1)) < 1 - \frac{C}{n}\right] \leq e^{-C}$, and (ii) there exist universal constants $c_0, c_1, c_2 > 0$ such that $\frac{c_1 \sqrt{\log n}}{n^{3/2}} \geq 1 - \mathbb{E}_{A,B}[V(x^*(c_0 \log n))] \geq 1 - \mathbb{E}_{A,B}[V(x^*)] \geq \frac{c_2 \sqrt{\log n}}{n^{3/2}}$.*

The theorem's first part shows Player 1 can attain a value arbitrarily close to 1 using a pure strategy, while the second part extends this to optimality up to constant factors with an $\mathcal{O}(\log n)$-sparse support. Its proof is constructive: concentrate weight on one action that already nearly maximizes Player 1's payoff, then mix in $\mathcal{O}(\log n)$ additional actions to compensate. Consequently, one can find an optimal SSE in expectation up to constant factors simply by using this explicit sparse strategy; no support-search is required.

**Random general-sum security games.** Given $T$ targets and $R$ identical defender resources, we consider a case when schedules form a partition of the $T$ targets[2]. That is, the resources can choose from $k$ non-empty schedules $S_1, \ldots, S_k \subseteq [T]$ which satisfy $\bigcup_{i \in [k]} S_i = [T]$ and $S_i \cap S_j = \emptyset$ for any $i, j \in [k]$ with $i \neq j$. We consider the following random model for utilities: $u_d^c(t) = u_a^c(t) = 0$ for all targets $t \in [T]$, while $u_d^u(t)$ and $u_a^u(t)$ are sampled i.i.d. uniformly in $[-1, 0]$ and $[0, 1]$, respectively. We expect our results to be relatively robust to the choice of distributions.

**Theorem 2.** *Let $u_d^u$ and $u_a^u$ be sampled i.i.d. uniformly in $[-1, 0]$ and $[0, 1]$. For $T$ targets and $R$ defender resources, let $\emptyset \subsetneq S_1, \ldots, S_k \subseteq [T]$, $1 \leq k < R$, form a partition of the targets, and denote $\alpha := \frac{\max_{i \in [k]} |S_i|}{\min_{i \in [k]} |S_i|}$. Then there exists a constant $c > 0$ such that $\mathbb{E}_u[V(x^*)] \geq -c \left( \sqrt{\frac{\alpha}{RT}} + \frac{1}{k} \right).$*

The proof is given in Appendix A.2. When $R \geq k$, we have $V(x^*) = 0$, and even if $R \in (0, 1]$, we need only use a single resource with probability at most $R$. In standard settings, the dominant error term is $-c\sqrt{\frac{\alpha}{RT}}$, with a smaller correction when schedules are few. Consequently, for random uniform security games, Player 1 achieves near-optimal utility with very few resources: if $\alpha = \mathcal{O}(1)$, then $\mathbb{E}[V(x^*)] \gtrsim -\mathcal{O}(1/\sqrt{RT})$, so a single resource already yields expected utility $-\mathcal{O}(1/\sqrt{T})$. Moreover, one can achieve near-maximum utility by using that resource only with probability $\mathcal{O}(1/\sqrt{T})$, a vanishingly small rate, a behavior unlikely in realistic settings.

## 4 Design of the Framework

The GUARD framework builds realistic security-game instances directly from publicly available, real-world data. As shown in Figure 1, the framework is organized into three core components: *Data*, *Games*, and *Solvers*. These components interact to define game instances from raw data inputs, model their structure and rules, and compute equilibrium solutions. Open-source datasets feed into a hierarchical game-class structure (Graph Game $\rightarrow$ Security Game $\rightarrow$ Domain-Specific Game), enabling users to instantiate realistic instances. Currently, GUARD supports three data streams–animal movement data, demographic data, and map data–as well as two domain-specific game implementations: Green Security Games (GSGs) and Infrastructure Security Games (ISGs). Users may construct games in either normal-form (NFG) or schedule-form (SFG), with all generation parameters detailed in C.3. In addition, we provide a library of preset, high-fidelity game instances to jump-start experimentation. Every game created within GUARD can be exported to standard formats (.pkl, .h5, and native *Gambit/Gamut* .nfg/.game files) and, conversely, imported via our Loaded Game class. While export functionality ensures compatibility with external libraries such as *Gambit* and *OpenSpiel*, GUARD also includes multiple built-in solvers for Nash and Stackelberg equilibria.

### 4.1 Data

Once processed, real-world input data populates the hierarchical game class structure, providing detailed parameter settings, target specifics, and environment constraints while preserving flexibility for user customization. Here we introduce the primary data sources used to instantiate realistic security games in the GUARD framework, while additional metrics drawn from real-world domain-specific literature are outlined in Section 5 and Appendices D.2, D.3, and D.4.

**Movebank.** To instantiate our GSGs, we utilize data from Movebank, a free, curated online platform for animal movement data maintained by the Max Planck Institute of Animal Behavior [49, 27]. Movebank hosts contributions from researchers globally and contains thousands of tracked individuals across diverse species and ecosystems. In our preset game instances, we aggregate nine GPS collar datasets for African Elephants from the *Elephant Research – Lobéké National Park (Cameroon)* study [32], which captures the movement of six unique elephants from 2002 to 2007. This dataset includes 3,183 spatiotemporal observations, and after preprocessing (Appendix D.1), we keep the following fields: animal id, latitude, longitude, and timestamp. These data form the basis for target generation and scoring in our game environment. Notably, any animal location dataset either from Movebank or another source when preprocessed can be used as input data for a GSG.

**OpenStreetMap.** To represent infrastructure targets in ISG instances, we extract data from Open-StreetMap (OSM) using Overpass Turbo [37], a query interface that supports structured data retrieval

---

[2]In particular, this includes also all simple schedules where every $S_i$ is a singleton.

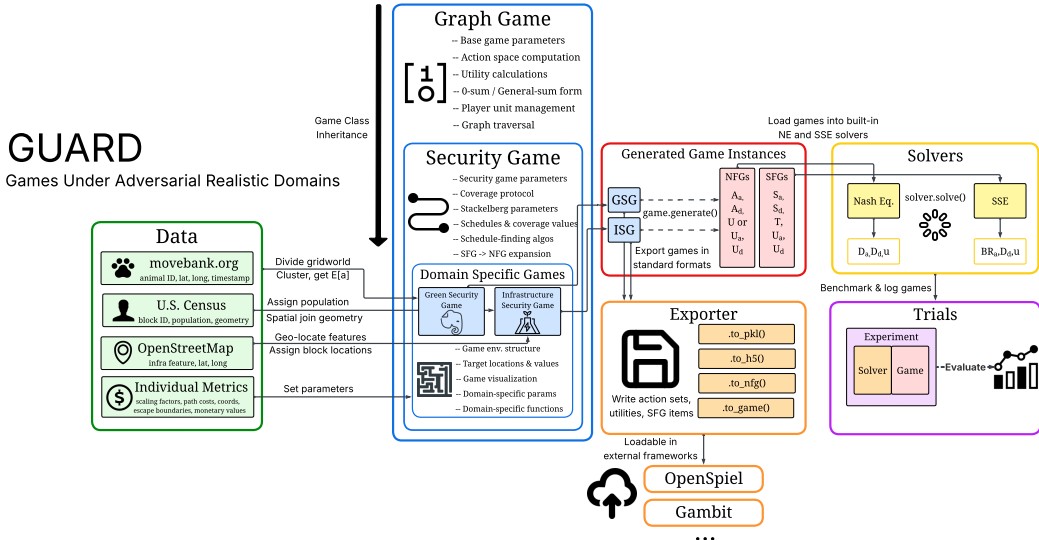

Figure 1: The structure of the GUARD framework.

from OSM's geospatial database (query in Appendix C.4). In our preset games, we query all power grid, medical, judicial, educational, and police features—including nodes (single coordinate locations, such as power stations or health clinics) and ways (sequences of nodes representing linear structures, such as power lines)—within the New York City metropolitan area. This region encompasses all five NYC boroughs and adjacent areas. Once filtered and preprocessed, the dataset includes 23,607 infrastructure features with attributes including ID, type, latitude, and longitude. These points serve as target locations in our game instances. This OSM data is interchangeable with any geographic region feature data for any desired game instance.

**Census.** In ISG instances, an attack on a given target must be assigned some value to both the defender and attacker. To do so, we use census data that aggregates regional population levels. Specifically, we utilize publicly available block-level population data from the 2020 U.S. Census TIGER/Line Shapefiles [47]. For our preset instances, the data provide both population counts and geospatial boundaries for 288,819 census blocks across the state of New York. Each entry includes a GEOID, population count, and a Polygon geometry specifying the spatial extent of the block.

## 4.2 Game Classes

GUARD follows a three-tiered game class hierarchy for flexible and extensible modeling. At the base, the *Graph Game* class defines core graph functionality, target definitions, action spaces, and utility matrices. The intermediate *Security Game* class adds domain-agnostic security mechanics, including defender/attacker movement, coverage rules, and NFG/SFG formulations. At the top, *Domain-Specific Game* classes integrate real-world data to define context-specific target locations and values, environment structure, and constraints, fully instantiating a real-world Security Game while leveraging the computational functionality of the lower layers.

**Graph Game.** The Graph Game class is a general-purpose abstraction for modeling dynamic multi-agent normal-form games on graphs, and provides a flexible, modular interface for generating data-driven security games. Users can configure the game graph, number and type of player resources (moving/stationary), game length, sum type (zero/general), and interdiction rules. Higher-level games are built by passing domain-specific parameters to this base class, which handles resource instantiation, action space generation, and (if specified) construction of NFG utility matrices according to zero or general sum specification. After computation, these action sets and utilities are then made available in the higher-level game classes for further specialization and experimentation. Specific parameterization details for the base Graph Game layer can be found in Appendix C.1.

The Graph Game initializes with a user-supplied directed graph $G = (V, E)$, which serves as the structural foundation for strategy computation. Path enumeration is performed via time-constrained

depth-first search (Appendix B.3), generating valid traversal-based action sets and enabling utility evaluation through node visitation and distance-based interdiction dynamics. In this base layer, the graph is time-expanded to support multi-timestep play. Thus, each player's action is a fixed-length 2D array encoding resource positions over $\mathcal{T}$ timesteps. For player $i \in \{a, d\}$ with $m_i$ resources, a full action is represented as a matrix $A^i = (v_{j,\tau})_{j \in [m_i], \tau \in [\mathcal{T}]} \in V^{m_i \times \mathcal{T}}$, where $v_{j,\tau}$ denotes the node occupied by resource $j$ at timestep $\tau$. In particular, stationary resources are encoded as constant rows in $A^i$. We define $\mathcal{A}_i$ as the set of all such possible actions for player $i$.

The utility for each player in a normal form representation is determined by the targets successfully captured by Player 2's attacking resources, subject to interdiction by Player 1's defending resources. The attacker's utility $U_a$ is defined as the sum of the values of all targets successfully captured:

$$U_a(A_a, A_d) = \sum_{t \in T} V_t \cdot \mathbb{1}\left[t \text{ is captured}\right] + C(A_a, A_d),$$

where $V_t$ is the value of target $t$, $\mathbb{1}[t \text{ is captured}]$ is the indicator of whether the target is successfully reached by an attacker without being interdicted, and $C(A_a, A_d)$ is an optional defender path cost term in general sum formulations, which can account for traversal costs or environmental factors. The Graph Game class encompasses various formulations for attacker resources being interdicted. Specifically, for an attacker *moving* resource $j_a \in [m_a]$, interdiction occurs if a defender resource $j_d \in [m_d]$ comes within the capture radius $r$ of the attacker:

$$\mathbb{1}\left[j_a \text{ is captured}\right] = \mathbb{1}[\exists j_d \in [m_d], \exists \tau \in \mathcal{T} : d(A^a_{j_a,\tau}, A^d_{j_d,\tau}) \le r],$$

where $d(v, w)$ is the distance between two nodes $v, w \in V$. For an attacker *stationary* resource $j_a \in [m_a]$ at target $t$, a defender resource must visit this target a minimum number of timesteps $\delta$ for the defender to prevent it from being captured, where $\delta$ is a defense time threshold:

$$\mathbb{1}[j_a \text{ is captured}] = \mathbb{1}[|\{\tau \in \mathcal{T} : \exists j_d \in [m_d] : A^a_{j_a,\tau} = t\}| \ge \delta].$$

**Security Game.** The Security Game layer extends the base Graph Game class by enforcing key constraints of standard security games: no stationary defenders, no moving attackers, and start/end constraints at home bases for defender paths. This creates a classic setting where mobile defender(s) patrol from fixed bases, and attacker(s) selects target(s) without traversing the graph.

This layer defines both zero-sum and general-sum formulations using a target utility matrix with covered and uncovered payoffs $u^c(t)$ and $u^u(t)$ for each target $t \in T$. The Security Game Layer also implements the Stackelberg schedule-form games when specified, passing SFG artifacts to the highest domain-specific layer for the user to access. These artifacts include defender resource schedule mappings, the target utility matrix, an expanded defender action set enumerating all possible defender schedule combinations, and NFG-expanded utility matrices. For general schedule-form games, this layer manages finding mappings from defenders to valid schedules which can be expanded into normal-form action spaces (and thus into the aforementioned NFG utility matrices for compatibility with broader classes of game-theoretic algorithms). See Appendix B.1 for schedule-finding algorithm implementations. Appendix C.2 details the specific parameters for the Security Game layer.

### 4.2.1 Domain Specific Games

**Green Security Games.** GSGs model the interaction between defenders (e.g., park rangers) and poachers in conservation areas as adversarial games over a spatial domain [20, 18]. The defender seeks to patrol animal location targets to establish coverage over animal targets and prevent poaching. Both simultaneous (pursuit-evasion) and Stackelberg formulations are supported. The game is played on a grid-world abstraction of a national park (nodes represent cells, edges connect adjacent cells).

Given spatiotemporal animal tracking data from Movebank or similar datasets, our framework infers target locations and values based on observation density. Users initialize a realistic GSG in the GUARD framework by supplying this data along with a bounding box, grid dimensions, a scoring method ("centroid" or "density"), and number of cluster targets if using the centroid method.

In the centroid method, K-means is applied to observation coordinates, and each cluster centroid is assigned a score proportional to its size, scaled by the animal-to-observation ratio: `score = cluster_size` $\times \left(\frac{\texttt{num\_animals}}{\texttt{num\_total\_observations}}\right)$. For general-sum games, the attacker value includes an optional escape proximity factor: `attacker_value = score` $\times$

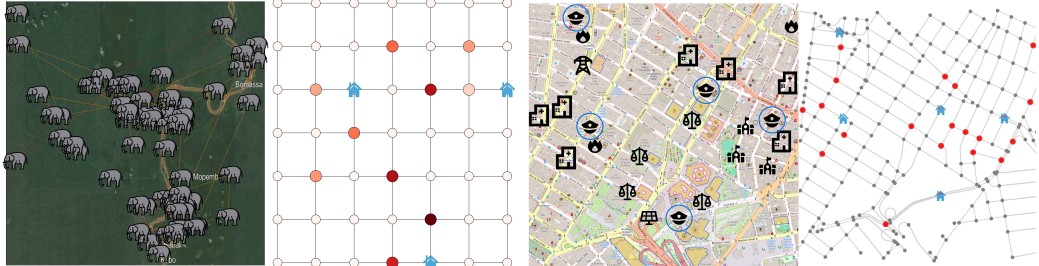

Figure 2: (Left) Down-sampled elephant movements in Lobéké National Park and the corresponding GSG model. (Right) Civil infrastructure in Manhattan's Chinatown and the corresponding ISG model. Graph structures represent the traversable game environments. Red nodes indicate target locations, blue house icons are home bases.

$\left(1 + \alpha \left(1 - \frac{d_i - d_{\min}}{d_{\max} - d_{\min}}\right)\right)$, `defender_value` $= -$`score`, where $d_i$ is the distance to the escape boundary, and $\alpha$ is the proximity scaling factor. In the density method, each observation contributes to the score of its containing cell, with values scaled by the overall animal-to-observation ratio.

**Infrastructure Security Games.** ISGs model adversarial interactions over civil infrastructure, where attackers aim to damage key assets (e.g., power grids, hospitals) and defenders patrol to protect and establish coverage over these sites. Like in GSGs, both simultaneous (pursuit-evasion) and Stackelberg formulations are supported. Games are typically played on a grid-like network representing an urban environment.

To initialize a realistic ISG in the GUARD framework, users supply: (i) a GeoPandas DataFrame of infrastructure features with coordinates and types; (ii) a block-level population GeoDataFrame; (iii) a weight dictionary for infrastructure types; (iv) a geographic bounding box; and (v) a population assignment method ("block" or "radius"). The block method assigns the population of the containing census block, while the radius method sums populations of intersecting blocks within a buffer around each feature. The framework then builds a street graph from OSM within the bounding box, maps infrastructure features to the nearest graph node, and computes each node's base score as `raw_score` $= W \cdot (\log(P + 1))^\alpha$, where $W$ is the infrastructure type weight, $P$ is the assigned population, and $\alpha$ is a scaling parameter controlling population importance.

In general-sum games, scores are adjusted for proximity to a predefined escape point: `attacker_value` $=$ `raw_score` $\times \left(1 + \alpha \left(\frac{d_{\max} - d_i}{d_{\max} - d_{\min}}\right)\right)$, `defender_value` $= -$`raw_score`, where $d_i$ is the distance to the escape point, and $\alpha$ controls proximity scaling. Final values are assigned to graph nodes to define the game's target set.

### 4.3 Solvers

The GUARD platform includes a suite of built-in solvers for computing Nash and Stackelberg equilibria across different security game instances. For zero-sum NFGs, we implement the standard Nash LP [42] and a support-bounded MILP [1] that fixes the size of one player's support. For iterative equilibrium computation, we include a range of no-regret and oracle-based methods. The no-regret methods include Regret Matching (RM) [24], Regret Matching+ (RM+) [45], and Predictive Regret Matching+ (PRM+) [21] (with each variant's parameter specifications discussed in Appendix D.3). For the oracle-based approaches, we provide both a standard Double Oracle algorithm [33] and its extension for schedule form; both are detailed in Appendix B.2. For general-sum games, we provide two algorithms. The simple SSE multiple LP [28] is used for security games with singleton schedules, while the general SSE multiple LP [13] is used for arbitrary NFGs.

### 4.4 Pre-defined Game Instances

GUARD comes loaded with several pre-defined realistic security game instances. These games are accessible as out-of-the-box .pkl files housing the entire pre-generated game objects for NFGs, and schedule form object dictionaries for SFGs. The suite of games includes GSGs for the aforementioend

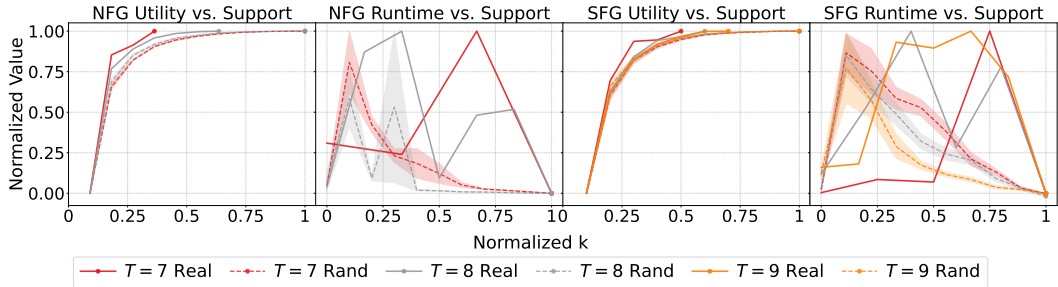

Figure 3: Comparison of GSG utility and runtime sparsity results for NFG and SFG settings with varying timesteps. Each subplot shows normalized utility and runtime as a function of normalized support, with real (solid) and random (dashed) lines for each timestep setting. Errorbars on random runs reflect standard error bounds ($\frac{\sigma}{\sqrt{n}}$) over 10 random seeds.

Lobéké National Park elephants in Cameroon and Etosha National Park elephants in Namibia [22]. It also includes ISGs for multiple locations with dense civil infrastructure in New York City. These settings are generated for 7-9 timestep games and available in both NFG and SFG format for zero-sum and general-sum formulations; full pre-defined game details are in Appendix C.8.

## 5   Empirical Evaluation

We compare equilibrium properties in selected typical realistic versus randomized games by evaluating (i-a) sparsity and runtime across support bounds and (i-b) convergence of iterative algorithms in zero-sum games, and (ii) defender utility, support size, and runtime in general-sum games. For GSGs, we use preset instances based on a $7 \times 7$ grid of Lobéké National Park, with 10 clustered elephant targets and 3 ranger bases, with a SFG target-coverage scaling ratio $u_d^u(t)/u_d^c(t)$ of 5 to reflect observed 80% poacher apprehension rates [9]. ISGs use Manhattan's Chinatown, with 23 infrastructure targets and 5 police stations as bases, with a SFG coverage scaling of 3 to match observed upper-end 67% urban crime mitigation rates [35, 26, 31].

**Sparsity Experiments.** We evaluate sparsity in zero-sum GSGs using 1 (NFG) or 2 (SFG) defenders and a defense time threshold of 1 (NFG) or 2 (SFG) timesteps, with randomized instances generated by uniformly sampling matrix entries within the observed range of real payoff values. Utility is normalized as $U_{\text{norm}} = \frac{U - U_{k=1}}{U_{\text{Nash}} - U_{k=1}}$, runtime as $R_{\text{norm}} = \frac{R - R_{\min}}{R_{\max} - R_{\min}}$, and support size as $k_{\text{norm}} = \frac{k}{k_{\text{max Nash}}}$ where $k_{\text{max Nash}}$ is the maximum Nash equilibrium support size across both real and randomized instances. Additional parameter specifications for these sparsity experiments are detailed in Appendix D.2. Results in Figure 3 show that real instances yield smaller supports at optimal utility, reaching approximately 40-60% (NFG) and 50-70% (SFG) of randomized support sizes across 7-, 8-, and 9-timestep games. Real instances also exhibit higher computational runtimes at larger supports, while randomized instances peak early and decline steadily.

**Iterative Algorithm Experiments.** In Figure 4, we evaluate the convergence behavior of four iterative algorithms—DO, RM, RM+, and PRM+—on both real and randomized instances in the GSG and ISG domains (with parameterization details in Appendix D.3), using NFG and SFG formulations. Across all algorithms, real instances consistently exhibit faster convergence than their randomized counterparts, particularly for the enhanced RM+ and PRM+ variants. In GSGs, real DO instances typically reach equilibrium faster, with smoother convergence trajectories. In ISGs, randomized DO often struggles to converge cleanly, exhibiting significant oscillations in utility gaps, especially in the NFG setting. These results indicate that real instances produce more stable convergence dynamics and reduced computational complexity, while randomized games introduce irregular behavior and inflated support sizes.

**Stackelberg Experiments.** We compute SSE for both simple (singleton) and general (multiple target) schedules in a general-sum formulation, comparing defender utility and support size across real and randomized GSG and ISG instances with parameterization details (including general sum path costs and proximity weights) in Appendix D.4. Randomized instances are generated by sampling target values uniformly within the range of real covered and uncovered target values, while ensuring covered

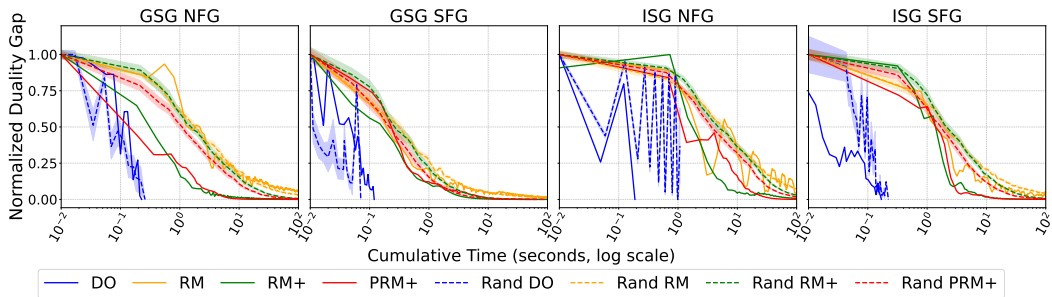

Figure 4: Convergence of iterative algorithms for different game types, formulations, and real vs. randomized (dashed lines with standard error bars) runs.

Table 1: SSE Results for GSG and ISG settings. RT = randomized target values, RM = randomized matrix, RTS = randomized target values and schedules. $u_d$ is max defender utility, **R** is runtime in seconds, and **S** is support of defender coverage strategy.

| Form | Trial | GSG | | | ISG | | |
|---|---|---|---|---|---|---|---|
| | | $u_d$ | **R** | **S** | $u_d$ | **R** | **S** |
| **Simple** | Real | -0.390 | 0.015s | 6 | -0.429 | 0.030s | 11 |
| | RT | $-0.298 \pm 0.03$ | $0.01s \pm 0$ | $5.0 \pm 0.52$ | $-0.214 \pm 0.05$ | $0.04s \pm 0$ | $7.5 \pm 1.20$ |
| **General** | Real | -0.207 | 0.46s | 9 | -0.408 | 838s | 14 |
| | RM | $-0.035 \pm 0$ | $0.52s \pm 0.02$ | $2 \pm 0.15$ | $-0.026 \pm 0$ | $541.5s \pm 1.98$ | $1.7 \pm 0.15$ |
| | RT | $-0.253 \pm 0$ | $0.35s \pm 0.01$ | $1.7 \pm 0.26$ | $-0.013 \pm 0$ | $116.2s \pm 1.87$ | $1 \pm 0$ |
| | RTS | $-0.270 \pm 0$ | $0.45s \pm 0.03$ | $1.4 \pm 0.16$ | $-0.032 \pm 0$ | $1.57 \pm 0.04$ | $1 \pm 0$ |

payoffs are strictly lower: $u_a^c(t) \le u_d^u(t)$, $\forall t$. Table 1 details our experimental results. With simple schedules, real instances tend to exhibit higher complexity, resulting in slightly larger support sizes and lower defender utilities compared to their randomized counterparts. The difference is more apparent with general schedules. We expand the schedule-form coverage strategy space into matrix-NFG form to run the general SSE multiple LP, and evaluate GUARD-generated realistic game instances against three randomized baselines: randomizing the expanded NFG matrices (RM), randomizing target values only (RT), and randomizing both target values and schedule assignments (RTS). Schedule randomization is detailed in Appendix D.5. In GSGs, the real instance yielded significantly larger support size than all randomized settings, which generally produced minimal supports (Real: $S = 9$, Rand: $S_{RM} = 2, S_{RT} = 1.7, S_{RTS} = 1.4$). Similarly, in ISGs, the real instance achieved substantially larger support and lower defender utilities compared to the randomized baselines which tend to inflate defender payoffs (Real: $u_d = -0.4076$, Rand: $u_d = -0.026, -0.013, -0.032$), and collapse to degenerate supports (Real: $S = 14$, Rand: $S_{RM} = 1.7, S_{RT} = 1, S_{RTS} = 1$). These observations are in close accord with theoretical results in Section 3: random general-sum instances often have degenerate support and give significant advantage to the defender, while realistic instances yield richer strategy profiles.

# 6 Conclusion

We present a flexible framework for generating structured games grounded in publicly available data, offering a range of customizable and preconfigured instances inspired by real-world security applications. Alongside this, we provide a dataset of ready-to-use games and support for exporting to standard formats. Our theoretical analysis highlights limitations of random games for evaluating equilibrium-finding algorithms, and our experiments confirm that such instances can exhibit degenerate behavior absent in more structured settings. Together, these contributions aim to support more robust and practical benchmarking in algorithmic game theory.

**Limitations.** In regards to limitations, GUARD faces scalability challenges in dense environments, where complex structure and rich data inflate game sizes and hinder algorithmic performance. Specifically, at high timestep settings, defender path action sets explode in size making utility matrices intractable to compute. More broadly, the framework depends on raw datasets that may imperfectly capture real-world dynamics, limiting fidelity to the underlying game environment. Also, our realistic security game instance generation was only tested in GSG and ISG domains. Additionally, some realistic instances exhibit degenerate behaviors as a consequence of certain environments having trivial best-response profiles under current target and action definitions. For example, data-driven targets may be unreachable for defender resources under certain game length and home-base constraints, causing trivial attacker best-responses and game degeneration. Finally, incorporating more real-world features like resource-specific subgraph constraints (modeling differing security patrol roles) could further enhance modeling realism. Limitations are discussed in greater detail in Appendix C.6.

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

# A  Proofs

## A.1  Proof of Theorem 1

First, we prove the first part of the theorem. Because all coordinates $B$ are uniform, with probability one, there are no two equal entries. We consider that this event holds in the rest of the proof. Then, for any $i \in [n]$, the best-response for the leader strategy $e_i$ is $y^*(e_i) = e_{j(i)}$ where $j(i) = \operatorname{argmax}_{j \in [n]} B_{i,j}$. Hence, since all entries of $B$ are i.i.d. the indices $j(i)$ for $i \in [n]$ are all i.i.d. uniform on $[n]$. In turn, since $A$ and $B$ are independent, the variables $e_i^\top A y^*(e_i) = A_{i,j(i)}$ for $i \in [n]$ are i.i.d. uniform on $[0,1]$. In summary, $V(x^*(1))$ is distributed as the maximum of $n$ i.i.d. uniform variables, that is, $V(x^*(1)) \sim \text{Beta}(n,1)$. The given expectation and deviation bounds are classical for this beta distribution.

Now we can move to the second part of the theorem. We prove each inequality separately, starting with the lower bound on the value of sparse solutions:

**Lemma 1.** *There exist universal constants $c_0, c_1 > 0$ such that for random uniform Stackelberg games,*

$$1 - \mathbb{E}[V(x^*(c_0 \log n))] \leq \frac{c_1 \sqrt{\log n}}{n\sqrt{n}}.$$

*Proof.* We give a constructive proof for this lower bound on $V(x^*(c_0 \log n))$, that is, we explicitly construct a $\mathcal{O}(\log n)$-sparse strategy for the leader then analyze its expected value. We did not optimize for constants in the following.

**Construction of the sparse strategy.** We first compute all pairs of indices for which the leader has the desired large payoff:

$$S := \{(i,j) : A_{i,j} \geq 1 - \delta_n\} \quad \text{where} \quad \delta_n := 2^{11} \frac{\sqrt{\log n}}{n\sqrt{n}}.$$

If $S = \emptyset$, we return any arbitrary strategy $x_{sparse} = e_1$. We suppose that $S \neq \emptyset$ from now. Next, we identify the pair of indices which yields the largest payoff for the follower, where ties can be broken arbitrarily:

$$(i^*, j^*) := \operatorname*{argmax}_{(i,j) \in S} B_{i,j}.$$

Next, we compute a set of other row indices. To do so, we start by identifying the rows $i$ for which $B_{i,j^*}$ is significant entries and the row $B_{i,\cdot}$ does not contain entries from $S$ other than possibly $(i, j^*)$,

$$R(j^*) := \left\{ i \in [n] \setminus \{i^*\} : B_{i,j^*} \geq \frac{3}{4}, \forall j \in [n] \setminus \{j^*\}, (i,j) \notin S \right\}.$$

Among these rows, we take the $c_0 \log n - 1$ rows maximizing the entries of $A_{\cdot,j^*}$. Precisely, we enumerate $R(j^*) = \{i_1, \ldots, i_{|R(j^*)|}\}$ where $A_{i_1,j^*} \geq A_{i_2,j^*} \geq \ldots \geq A_{i_{|R(j^*)|},j^*}$. We then pose

$$R^* := \{i_s, s \leq c_0 \log n - 1\} \quad \text{where} \quad c_0 = 200.$$

The final leader strategy that we use is

$$x_{sparse} := (1 - \eta_n)e_{i^*} + \frac{\eta_n}{|R^*|} \sum_{i \in R^*} e_i, \quad \text{where} \quad \eta_n := 8(1 - B_{i^*,j^*}).$$

**Analysis of the sparse strategy value.** The main steps of the analysis are to show that the best response for the follower is $y^*(x_{sparse}) = e_{j^*}$, and that the leader has large value for $x_{sparse}^\top A e_{j^*}$.

First, because all entries of $A$ are i.i.d. uniform on $[0,1]$, the matrix $(\mathbb{1}[A_{i,j} \geq 1 - \delta_n])_{i,j}$ has i.i.d. Bernoulli entries $\text{Ber}(\delta_n)$. In particular, $|S| \sim \text{Binom}(n^2, \delta_n)$. Using Chernoff's bound, we have

$$\mathbb{P}\left[\left||S| - n^2\delta_n\right| > \frac{1}{2}n^2\delta_n\right] \leq 2e^{-\frac{1}{12}n^2\delta_n}. \tag{1}$$

For convenience, we denote by $\mathcal{E} = \left\{\frac{1}{2}n^2\delta_n \leq |S| \leq \frac{3}{2}n^2\delta_n\right\}$ the complementary event. Conditionally on $S$, $B$ still has all entries i.i.d. uniform on $[0,1]$. Then, conditional on $S$ and $S \neq \emptyset$,

$B_{i^*,j^*} \sim \text{Beta}(S, 1)$ is distributed as the maximum of $S$ i.i.d. uniforms on $[0, 1]$. In particular, we have

$$\mathbb{E}\left[B_{i^*,j^*} \mid S, \mathcal{E}\right] = \frac{S}{S+1} \geq 1 - \frac{2}{n^2 \delta_n}, \tag{2}$$

where in the last inequality we used that fact that on $\mathcal{E}$, $|S| \geq \frac{1}{2}n^2 \delta_n$.

In the rest of the proof, we reason conditionally on $S$ and $\{B_{i,j}, (i,j) \in S\}$. Conditionally to these, the other coordinates $A_{i,j}$ and $B_{i,j}$ for $(i,j) \notin S$ are still all independent and distributed as uniform variables on $[0, 1 - \delta_n]$ for $A_{i,j}$ and uniforms on $[0, 1]$ for $B_{i,j}$. To analyze $R(j^*)$ we first introduce the following sets of rows:

$$R_0 := \{i \in [n] : \exists j \in [n], (i,j) \in S\}$$
$$R_1(j^*) := R(j^*) \cap R_0 = \{i \in [n] \setminus \{i^*\} : (i, j^*) \in S, \forall j \in [n] \setminus \{j^*\}, (i,j) \notin S\}.$$

Clearly, we have $|R_0| \leq |S|$ and $i^* \in R_0$ since $(i^*, j^*) \in S$. Conditionally on $S$ and $\{B_{i,j}, (i,j) \in S\}$ the entries $B_{i,j^*}$ for all $i \notin R_0$ are distributed as i.i.d. uniforms on $[0, 1]$. Then, $R(j^*) \setminus R_0$ is distributed as independently including each element from $[n] \setminus R_0$ with probability $1/4$. In particular, conditional on $S$ and $\{B_{i,j}, (i,j) \in S\}$,

$$|R(j^*) \setminus R_0| \sim \text{Binom}\left(n - |R_0|, \frac{1}{4}\right).$$

Note that under $\mathcal{E}$, we have $n - |R_0| \geq n - |S| \geq n - \frac{3}{2}n^2 \delta_n \geq \frac{n}{2}$ for $n$ sufficiently large. Under that event, $|R(j^*) \setminus R_0|$ therefore stochastically dominates a binomial $\text{Binom}(n/2, 1/4)$. Hence, Chernoff's bound implies

$$\mathbb{P}\left[|R(j^*) \setminus R_0| \leq \frac{n}{16} \mid S, j^*, \mathcal{E}\right] \leq e^{-n/64}. \tag{3}$$

For convenience, we denote $\mathcal{G} := \{|R(j^*) \setminus R_0| \geq n/16\}$. In particular, for $n$ sufficiently large, under $\mathcal{G}$, $R(j^*)$ contains at least $l_0 := \lfloor c_0 \log n \rfloor - 1$ row indices. Hence under $\mathcal{G}$ we have $|R^*| = l_0$.

We next compute

$$\sum_{i \in R^*} A_{i,j^*} = \sum_{i \in R^* \cap R_1(j^*)} A_{i,j^*} + \sum_{i \in R^* \setminus R_0} A_{i,j^*}$$

$$\overset{(i)}{\geq} |R^* \cap R_1(j^*)|(1 - \delta_n) + \sum_{i \in R^* \setminus R_0} A_{i,j^*}, \tag{4}$$

where in $(i)$ we used the definition of $R_1(j^*)$ implying that any $i \in R_1(j^*)$ satisfies $(i, j^*) \in S$. Recall that within $R^*$ we include the first $l_0 := \lfloor c_0 \log n \rfloor - 1$ rows by decreasing order of $A_{i,j^*}$ for $j^* \in R(j^*)$. Therefore, $R^* \setminus R_0$ corresponds to the first $|R^* \setminus R_0|$ rows by decreasing order of $A_{i,j^*}$ among $R(j^*) \setminus R_0$. Recall that conditionally on $S$ and $\{B_{i,j}, (i,j) \in S\}$, all entries $A_{i,j^*}$ and $B_{i,j^*}$ for $i \notin R_0$ are independent. As a result, conditionally on $S$, $\{B_{i,j}, (i,j) \in S\}$, and $R(j^*)$, all entries $A_{i,j^*}$ for $i \in R(j^*) \setminus R_0$ are still distributed as i.i.d. uniforms on $[0, 1 - \delta_n]$. For convenience, let us define $l_1 := l_0 - |R^* \cap R_1(j^*)|$ and $l_2 := |R(j^*) \setminus R_0|$. In summary, conditionally on $S$, $R(j^*)$, $R_0$, $R_1(j^*)$, and $\mathcal{G}$,

$$\sum_{i \in R^* \setminus R_0} A_{i,j^*} \sim (1 - \delta_n) \sum_{i=0}^{l_1 - 1} X_{(l_2 - i : l_2)},$$

where $(Y_i)_{i \in [l_2]}$ is an i.i.d. sequence of uniforms on $[0, 1]$, and $Y_{(a:n)}$ denotes the $a$-th order statistic of $(Y_i)_{i \in [l_2]}$. Using standard results on the expectation of order statistics of uniforms, we can further the bounds from Eq. (4) to obtain

$$\mathbb{E}\left[\sum_{i \in R^*} A_{i,j^*} \mid S, R(j^*), R_0, R_1(j^*), \mathcal{G}\right] \overset{(i)}{\geq} (1 - \delta_n)(l_0 - l_1) + (1 - \delta_n) \sum_{i=0}^{l_1 - 1} \frac{l_2 - i}{l_2 + 1}$$

$$= (1 - \delta_n)l_0 - (1 - \delta_n)\frac{l_1(l_1 + 1)}{l_2 + 1}$$

$$\overset{(ii)}{\geq} (1 - \delta_n)\left(1 - \frac{l_0 + 1}{l_2 + 1}\right) l_0$$

$$\overset{(iii)}{\geq} \left(1 - \delta_n - 16c_0 \frac{\log n}{n}\right)|R^*|.$$

In $(i)$ we used $\mathbb{E}[X_{(a:b)}] = \frac{a}{b+1}$ for the expectation of the $a$-th order statistics for $b$ i.i.d. uniforms on $[0,1]$. In $(ii)$ we used $l_1 \leq l_0$. In $(iii)$ we used $l_2 \geq n/16$ and $|R^*| = l_0$ since $\mathcal{G}$ is satisfied. As a summary, for $n$ sufficiently large, we obtained

$$\mathbb{E}\left[\frac{1}{R^*}\sum_{i\in R^*} A_{i,j^*} \mid S, R(j^*), R_0, R_1(j^*), B_{i^*,j^*}, \mathcal{G}\right] \geq 1 - 17c_0\frac{\log n}{n}. \tag{5}$$

By construction of $(i^*, j^*)$, we also have $A_{i^*,j^*} \geq 1 - \delta_n$. Together with the previous equation this intuitively shows that $x_{sparse}^\top Ae_{j^*}$ has large value for the leader. Precisely, using Eq. (5), we have

$$\mathbb{E}\left[x_{sparse}^\top Ae_{j^*} \mid S, R(j^*), R_0, R_1(j^*), B_{i^*,j^*}, \mathcal{G}\right] = (1-\eta_n)A_{i^*,j^*} + \eta_n\left(1 - 17c_0\frac{\log n}{n}\right)$$

$$\geq 1 - \delta_n - 17c_0\frac{\log n}{n}\eta_n. \tag{6}$$

We now show that $e_{j^*}$ is indeed the best response of the follower for $x_{sparse}$. Equivalently, we need to show that the vector $x_{sparse}^\top B = (1-\eta_n)B_{i^*,\cdot} + \frac{\eta_n}{|R^*|}\sum_{i\in R^*} B_{i,\cdot}$ attains its maximum for column $j^*$. To do so, we introduce the following event

$$\mathcal{H} := \left\{\max_{j\in[n]\setminus\{j^*\}} \frac{1}{|R^*|}\sum_{i\in R^*} B_{i,j} - \frac{1}{2} < \frac{1}{8}\right\}.$$

By construction of $R(j^*)$, we know that for all $i \neq i^*$ and $j \neq j^*$, one has $(i,j) \notin S$. In particular, conditionally on $S$, $\{B_{i,j}, (i,j) \in S\}$, $\{B_{i,j^*}, i \in [n]\}$, and $R(j^*)$, all entries $B_{i,j}$ for $i \in R(j^*)$ and $j \neq j^*$ are still i.i.d. uniform on $[0,1]$. Next, constructing $R^*$ from $R(j^*)$ only involves the quantities $A_{i,j^*}$ for $i \in R(j^*)$, which are independent from the entries in $B$. Hence, conditionally on $\mathcal{I} := \{S, \{B_{i,j}, (i,j) \in S\}, \{B_{i,j^*}, i \in [n]\}, R(j^*), R_0, R_1(j^*), R^*, \mathcal{G}\}$, the matrix $(B_{i,j})_{i\in R^*, j\in[n]\setminus\{j^*\}}$ is exactly distributed as a $l_0 \times (n-1)$ random matrix with all entries i.i.d. uniform on $[0,1]$. Therefore, applying Hoeffding's bound on each column $j \in [n] \setminus \{j^*\}$ then the union bound over all these columns, we have

$$\mathbb{P}[\mathcal{H}^c \mid \mathcal{I}] \leq (n-1)e^{-|R^*|/32} = (n-1)e^{-l_0/32}. \tag{7}$$

In the last equality we used $|R^*| = l_0$ under $\mathcal{G}$.

Suppose that $\mathcal{H}$ holds. Then, one one hand,

$$1 - (x_{sparse}B)_{j^*} = (1-\eta_n)(1-B_{i^*,j^*}) + \frac{\eta_n}{|R^*|}\sum_{i\in R^*}(1-B_{i,j^*}) \overset{(i)}{\leq} (1-B_{i^*,j^*}) + \frac{\eta_n}{4} \overset{(ii)}{=} \frac{3\eta_n}{8},$$

where in $(i)$ we used $R^* \subset R(j^*)$ and the fact that $\mathcal{F}$ holds. In $(ii)$ we used the definition of $\eta_n$. On the other hand,

$$\min_{j\in[n]\setminus\{j^*\}} 1 - (x_{sparse}B)_{j^*} \geq \eta_n \cdot \min_{j\in[n]\setminus\{j^*\}}\left(1 - \frac{1}{|R^*|}\sum_{i\in R^*} B_{i,j}\right) \overset{(i)}{>} \frac{3\eta_n}{8},$$

where in $(i)$ we used the fact that $\mathcal{H}$ holds. In summary, we obtained

$$\mathcal{H} \subseteq \{y^*(x_{sparse}) = e_{j^*}\}. \tag{8}$$

where in the last inequality we used Eq. (7). Putting everything together, we obtained

$$1 - \mathbb{E}[V(x_{sparse})]$$
$$\leq \mathbb{P}[(\mathcal{E} \cap \mathcal{G})^c] + \mathbb{E}[(1 - V(x_{sparse}))\mathbb{1}[\mathcal{E}]\mathbb{1}[\mathcal{G}]]$$
$$\leq \mathbb{P}[\mathcal{E}^c] + \mathbb{P}[\mathcal{G}^c \mid \mathcal{E}] + \mathbb{P}[y^*(x_{sparse}) \neq e_{j^*} \mid \mathcal{E}, \mathcal{G}] + \mathbb{E}[(1 - x_{sparse}^\top Ae_{j^*})\mathbb{1}[\mathcal{E}]\mathbb{1}[\mathcal{G}]]$$
$$\overset{(i)}{\leq} \mathbb{P}[\mathcal{E}^c] + \mathbb{P}[\mathcal{G}^c \mid \mathcal{E}] + \mathbb{P}[\mathcal{H}^c \mid \mathcal{E}, \mathcal{G}] + \delta_n + 2^{12}c_0\frac{\log n}{n} \cdot \mathbb{E}[(1 - B_{i^*,j^*})\mathbb{1}[\mathcal{E}]\mathbb{1}[\mathcal{G}]]$$
$$\leq \mathbb{P}[\mathcal{E}^c] + \mathbb{P}[\mathcal{G}^c \mid \mathcal{E}] + \mathbb{P}[\mathcal{H}^c \mid \mathcal{E}, \mathcal{G}] + \delta_n + 2^{12}c_0\frac{\log n}{n} \cdot \mathbb{E}[1 - B_{i^*,j^*} \mid \mathcal{E}]$$
$$\overset{(ii)}{\leq} n^{1-\frac{c_0}{64}} + \delta_n + 2^{13}c_0\frac{\log n}{n^3\delta_n} \leq \frac{1}{n^2} + 2\delta_n.$$

In $(i)$ we used Eq. (8) and the bound from Eq. (6) together with the definition of $\eta_n$. In $(ii)$ we used Eqs. (1) to (3) and (7), and took $n$ large enough so that the term $\mathbb{P}[\mathcal{H}^c \mid \mathcal{E}, \mathcal{G}]$ from Eq. (7) dominates and $l_0 \geq \frac{c_0}{2} \log n$. Because $x_{sparse}$ only has $l_0 + 1 \leq c_0 \log n$ non-zero entries, this ends the proof that

$$1 - \mathbb{E}[V(x^*(c_0 \log n))] \leq 1 - \mathbb{E}[V(x_{sparse})] \leq c_1 \frac{\sqrt{\log n}}{n\sqrt{n}},$$

for some universal constant $c_1 \geq 2$. □

We next turn to the upper bound on the value of the Stackelberg game $\mathbb{E}[V(x^*)]$.

**Lemma 2.** *There exists a universal constants $c_2 > 0$ such that for random uniform Stackelberg games,*

$$1 - \mathbb{E}[V(x^*)] \geq \frac{c_2 \sqrt{\log n}}{n\sqrt{n}}.$$

*Proof.* We will use similar notations as in the proof of the lower bound for $\mathbb{E}[V(x_{sparse})]$ in Lemma 1. We define

$$S := \{(i,j) : A_{i,j} \geq 1 - \epsilon_n\} \quad \text{where} \quad \epsilon_n := \frac{c\sqrt{\log n}}{n\sqrt{n}},$$

where $c > 0$ is a constant to be fixed. As in the proof of Lemma 1, the distribution of $S$ corresponds to adding each entry $(i,j)$ independently with probability $\epsilon_n$. Hence, Chernoff's bound implies that

$$\mathbb{P}\left[\left||S| - n^2\epsilon_n\right| > \frac{1}{2}n^2\epsilon_n\right] \leq 2e^{-\frac{1}{12}n^2\epsilon_n} = 2e^{-\frac{c}{12}\sqrt{n}}. \tag{9}$$

We define the complementary event by $\mathcal{E} := \{\frac{c}{2}\sqrt{n\log n} \leq |S| \leq \frac{3c}{2}\sqrt{n\log n}\}$. Next, we introduce the event that there are at most $\log n$ columns that contain at least 2 elements of $S$:

$$\mathcal{F}_1 := \{|\{j \in [n] : \exists i_1 \neq i_2 \in [n], (i_1, j), (i_2, j) \in S\}| \leq \log n\}.$$

We have

$$\mathbb{P}[\mathcal{F}_1^c] \leq \frac{1}{\log n} \sum_{j=1}^{n} \sum_{i_1 \neq i_2 \in [n]} \mathbb{1}[(i_1, j), (i_2, j) \in S] \leq \frac{n^3 \epsilon^2}{\log n} = c^2. \tag{10}$$

In the first inequality we used Markov's inequality. We next define the event in which $S$ does not contain three entries in the same column:

$$\mathcal{F}_2 := \{\forall j \in [n] : |\{i \in [n] : (i,j) \in S\}| \leq 3\}.$$

We give a lower bound on the probability of this event as follows,

$$\mathbb{P}[\mathcal{F}_2^c] \leq \mathbb{E}\left[\sum_{i \in [n]} \sum_{j_1, j_2, j_3 \in [n] \text{ distinct}} \mathbb{1}[(i, j_1), (i, j_2), (i, j_3) \in S]\right] \leq n^4 \cdot \epsilon_n^3 \leq c^3 \frac{\log^2 n}{\sqrt{n}}. \tag{11}$$

Last, we define the event $\mathcal{F} := \mathcal{F}_1 \cap \mathcal{F}_2$. We suppose that this event is met in the rest of the proof. We define $\mathcal{G}$ the set of good columns which contain exactly one element of $S$ and $\mathcal{B}$ the set of bad columns containing at least 2 elements of $S$. Under $\mathcal{F}_1$ we have $|\mathcal{B}| \leq \log n$ and all columns of $\mathcal{B}$ contain exactly 2 elements of $S$. For any good column $j \in \mathcal{G}$ we let $i(j) \in [n]$ be the row for which $(i(j), j) \in S$. In the proof, we will treat separately columns $\mathcal{G}$ and $\mathcal{B}$. More precisely, consider the optimal strategy for the leader $x^\star \in \Delta^n$. Up to resolving ties in favor of the leader, there exists a column $j \in [n]$ for which $x^\star A e_j \geq V(x^\star)$. Now note that if $j \notin \mathcal{G} \cup \mathcal{B}$, we have for any $x \in \Delta^n$:

$$1 - xAe_j = \sum_{i \in [n]} x_i(1 - A_{i,j}) \geq \epsilon_n,$$

since for all $i \in [n]$, $(i,j) \notin S$. In other terms, for any $\eta < \epsilon_n$,

$$\{V(x^\star) > 1 - \eta\} \subseteq \bigcup_{j \in \mathcal{G} \cup \mathcal{B}} \left\{\exists x \in \Delta^n, j = \underset{j' \in [n]}{\operatorname{argmax}}(x^\top B)_{j'}, x^\top A e_j > 1 - \eta\right\}. \tag{12}$$

From now, we reason conditionally on $\mathcal{I} := \{S, \mathcal{E}, \mathcal{F}, \{A_{i,j}, (i,j) \in S\}\}$. In particular, unless mentioned otherwise, we always assume that $\mathcal{E}$ and $\mathcal{F}$ are satisfied. Under this conditioning, the other entries of $A$ and $B$ are still all independent: the entries $A_{i,j}$ for $(i,j) \notin S$ are i.i.d. uniform in $[0, 1 - \epsilon_n]$ while $B$ has all entries i.i.d. uniform in $[0, 1]$. Note that conditionally on $\mathcal{I}$, the variables $B_{i(j),j}$ for $j \in \mathcal{G}$ are i.i.d. uniform on $[0, 1]$. Then, we order $\mathcal{G} = \{j_s, s \in [|\mathcal{G}|]\}$ by decreasing order of $B_{i(j_s),j_s}$ for $j_s \in [|\mathcal{G}|]$. For convenience, we write $i_s := i(j_s)$. From the previous discussion, conditionally on $\mathcal{I}$, the variables $(B_{i_s,j_s})_{s \in [|\mathcal{G}|]}$ are distributed as the order statistics of $|\mathcal{G}|$ i.i.d. uniforms on $[0, 1]$ (in reverse order). We next give bounds on the quantities

$$\Delta_s := 1 - B_{i_s,j_s}, \quad s \in [|\mathcal{G}|].$$

From the previous discussion, for any $s \le |\mathcal{G}|/4$ and $\alpha \ge 10$, we have

$$\mathbb{P}\left[\Delta_s \le \frac{s}{\alpha|\mathcal{G}|} \mid \mathcal{I}\right] = \mathbb{P}_{Y \sim \text{Binom}(|\mathcal{G}|, \frac{s}{\alpha|\mathcal{G}|})}[Y \ge s]$$

$$\overset{(i)}{\le} e^{-|\mathcal{G}|D\left(\frac{s}{|\mathcal{G}|} \| \frac{s}{\alpha|\mathcal{G}|}\right)} \overset{(ii)}{\le} \frac{1}{(\alpha - 1)^{s(1-1/\alpha)}} \le (2/\alpha)^{s/2}.$$

where in $(i)$ the last inequality, we used Chernoff's bound and in $(ii)$ we used the identity $D(p + \epsilon \| p) \ge \frac{\epsilon}{2} \ln \frac{\epsilon}{p}$ whenever $\epsilon \ge 8p$ and $p + \epsilon \le 1/4$ (e.g. see Lemma 16 from [4]). When $s \ge |\mathcal{G}|/4$, we can directly use

$$\mathbb{P}\left[\Delta_s \le \frac{s}{2|\mathcal{G}|} \mid \mathcal{I}\right] \le e^{-|\mathcal{G}|D\left(\frac{s}{|\mathcal{G}|} \| \frac{s}{2|\mathcal{G}|}\right)} \le e^{-c_1|\mathcal{G}|} \le e^{-c_1\left(\frac{c}{2}\sqrt{n \log n} - 3\log n\right)} \le e^{-c_2\sqrt{n}},$$

for some universal constants $c_1, c_2 > 0$. In the last inequality, we used the event $\mathcal{E}$ and $\mathcal{F}$, under which we have $|\mathcal{G}| \ge |S| - 3|\mathcal{B}| \ge \frac{c}{2}\sqrt{n \log n} - 3 \log n$. We then define the event

$$\mathcal{G}(\alpha) := \bigcap_{s \in [S]} \left\{\Delta_s \ge \frac{s}{2\alpha c\sqrt{n \log n}}\right\}.$$

Since we have $|\mathcal{G}| \le |S| \le 2c\sqrt{n \log n}$ on $\mathcal{E}$, the previous tail bounds imply for any $s \in [|\mathcal{G}|]$,

$$\mathbb{P}[\mathcal{G}(\alpha)^c \mid \mathcal{I}] \ge \sum_{s \ge 1}(2/\alpha)^{s/2} + ne^{-c_2\sqrt{n}} \le \frac{c_3}{\sqrt{\alpha}} + e^{-c_3\sqrt{n}}, \tag{13}$$

for some universal constant $c_3 > 0$. We next add the variables $B_{i,j}$ for $(i,j) \in S$ to the conditioning $\mathcal{J} := \mathcal{I} \cup \{B_{i,j}, (i,j) \in S\}$, which does not affect the distribution of $A_{i,j}$ and $B_{i,j}$ for $(i,j) \notin S$.

We now fix $s \in [|\mathcal{G}|]$. We denote by $i_s^{(l)}$ the index $i \in [n] \setminus \{i_s\}$ with the $l$-th largest value of $A_{i,j_s}$. We define for $l \in [n-1]$ the event

$$\mathcal{H}_s^1(\alpha; l) := \left\{\exists j \in [n] \setminus (\mathcal{G} \cup \mathcal{B}) : 1 - B_{i_s,j} \le \frac{s}{4\alpha c\sqrt{n \log n}} \text{ and } \forall l' \in [l] : B_{i_s^{(l')},j} \ge B_{i_s^{(l')},j_s}\right\}.$$

We recall that $|\mathcal{G} \cup \mathcal{B}| \le |S| \le 2c\sqrt{n \log n}$ under $\mathcal{E}$. In particular, for $n$ sufficiently large we have $|[n] \setminus (\mathcal{G} \cup \mathcal{B})| \ge n/2$. Also, conditionally on $\mathcal{J}$, all values of $B_{i,j_s}$ for $i \ne i_s$, and $B_{i,j}$ for $j \notin \mathcal{G} \cup \mathcal{B}$ are i.i.d. uniform on $[0, 1]$. Hence, we can directly bound the probability of failure of $\mathcal{H}_s^1(\alpha; l)$ as follows:

$$\mathbb{P}[\mathcal{H}_s^1(\alpha; l)^c \mid \mathcal{J}] = \mathbb{E}_{B_{i_s^{(1)},j_s}, \ldots, B_{i_s^{(l)},j_s}}\left[\left(1 - \frac{s(1 - B_{i_s^{(1)},j_s}) \ldots (1 - B_{i_s^{(l)},j_s})}{4\alpha c\sqrt{n \log n}}\right)^{n - |\mathcal{G} \cup \mathcal{B}|}\right]$$

$$\le \mathbb{E}_{B_{i_s^{(1)},j_s}, \ldots, B_{i_s^{(l)},j_s}}\left[\exp\left(-\frac{s(1 - B_{i_s^{(1)},j_s}) \ldots (1 - B_{i_s^{(l)},j_s})}{8\alpha c}\sqrt{\frac{n}{\log n}}\right)\right].$$

We recall that all the variables $B_{i_s^{(1)},j_s}, \ldots, B_{i_s^{(l)},j_s}$ are i.i.d. uniform on $[0, 1]$. In particular, for $l = 1$, we have

$$\mathbb{P}[\mathcal{H}_s^1(\alpha; 1)^c \mid \mathcal{J}] \le \frac{1}{n^2} + 16\alpha c\frac{(\log n)^{3/2}}{s\sqrt{n}},$$

where we distinguished whether $1 - B_{i'_s,j_s} > 16\alpha c(\log n)^{3/2}/(s\sqrt{n})$ or not. Next, for $l \geq 2$, $-\log(1 - B_{i_s^{(1)},j_s}), \ldots, -\log(1 - B_{i_s^{(l)},j_s})$ are i.i.d. exponential $\mathcal{E}(1)$ variables. Hence, $(1 - B_{i_s^{(1)},j_s}) \ldots (1 - B_{i_s^{(l)},j_s})$ is distributed as $e^{-Z}$ were $Z \sim \text{Erlang}(l,1)$. Then, letting $l_n := \lfloor \frac{1}{8}\log n \rfloor$ and considering whether $Z \leq 2l_n$ or not,

$$\mathbb{P}[\mathcal{H}_s^1(\alpha; l_n)^c \mid \mathcal{J}] \leq \exp\left(-\frac{e^{-2l_n}\sqrt{n}}{8\alpha c\sqrt{\log n}}\right) + \mathbb{P}[Z \geq 2l_n]$$

$$\overset{(i)}{\leq} \exp\left(-\frac{n^{1/4}}{8\alpha c\sqrt{\log n}}\right) + 2^{l_n}e^{-2l_n} \leq \exp\left(-\frac{n^{1/4}}{8\alpha c\sqrt{\log n}}\right) + \frac{c_4}{n^{c_4}},$$

for some universal constant $c_4 > 0$. In $(i)$, we used $\mathbb{P}[Z \geq x] = e^{-x}\sum_{n=0}^{l_n-1}\frac{1}{n!}x^n$ for $x \geq 0$. We then define the event

$$\mathcal{H}^1(\alpha) := \bigcap_{s \in [|\mathcal{G}|], s \geq (\log n)^3} \mathcal{H}_s^1(\alpha; 1) \cap \bigcap_{s \in [|\mathcal{G}|], s \leq (\log n)^3} \mathcal{H}_s^1(\alpha; l_n).$$

From now, we suppose that $\alpha \leq n^{1/8}$. Then, taking the union bound, we obtained

$$\mathbb{P}[\mathcal{H}^1(\alpha)^c \mid \mathcal{J}] \leq \frac{c_5}{n^{c_5}}, \tag{14}$$

for some universal constant $c_5 > 0$.

Next, we recall that conditionally on $\mathcal{J}$ all these variables $A_{i,j_s}$ for $i \neq i_s$ are i.i.d. and stochastically dominated by the uniform distribution on $[0,1]$. We introduce the following event for $l < n - 1$

$$\mathcal{H}_s^2(\alpha; l) := \left\{1 - A_{i_s^{(l+1)},j_s} \geq \frac{l+1}{\alpha n s^{2/3}}\right\}.$$

If $l = 1$, we can bound its probability of failure as follows

$$\mathbb{P}[\mathcal{H}_s^2(\alpha; 1)^c \mid \mathcal{J}] \leq n^2\left(\frac{2}{\alpha n s^{2/3}}\right)^2 = \frac{4}{\alpha^2 s^{4/3}}.$$

In the first inequality, we used the fact that by construction, $A_{i_s^{(2)},j_s}$ is the second largest among $A_{i,j_s}$ for $i \neq i_s$, and hence is stochastically dominated by the second largest value among $n$ i.i.d. uniforms on $[0,1]$. Similarly, recalling that $l_n = \lfloor \frac{1}{8}\log n \rfloor$, and $\alpha \geq 10$, we can also bound the probability of failure for $l = l_n$ as follows:

$$\mathbb{P}[\mathcal{H}_s^2(\alpha; l_n)^c \mid \mathcal{J}] \leq \mathbb{P}_{Y \sim \text{Binom}(n-1, \frac{l_n+1}{\alpha n s^{2/3}})}[Y \geq l_n + 1]$$

$$\overset{(i)}{\leq} e^{-(n-1)D(\frac{l_n+1}{n-1}\|\frac{l_n+1}{\alpha n})} \leq \left(\frac{2}{\alpha}\right)^{\frac{l_n+1}{4}} \leq \frac{c_6}{n^{c_6}},$$

for some universal constant $c_6 > 0$. In summary, letting

$$\mathcal{H}^2(\alpha) := \bigcap_{s \in [|\mathcal{G}|], s \geq (\log n)^3} \mathcal{H}_s^2(\alpha; 1) \cap \bigcap_{s \in [|\mathcal{G}|], s \leq (\log n)^3} \mathcal{H}_s^2(\alpha; l_n),$$

taking the union bound, we also obtained

$$\mathbb{P}[\mathcal{H}^2(\alpha)^c \mid \mathcal{J}] \leq \frac{c_7}{\alpha^2} + \frac{c_7}{n^{c_7}}, \tag{15}$$

for some universal constant $c_7 > 0$. With all these events at hand, we are ready to treat each column $j \in \mathcal{G}$ separately. In the following, we suppose that $\mathcal{E}, \mathcal{F}, \mathcal{G}(\alpha), \mathcal{H}^1(\alpha), \mathcal{H}^2(\alpha)$ hold and that $10 \leq \alpha \leq n^{1/8}$. We fix $s \in [|\mathcal{G}|]$ and for convenience, we write $l_n(s) := l_n$ if $s \leq (\log n)^3$ and otherwise $l_n(s) := 1$.

Let $x \in \Delta^n$ such that $j_s \in \text{argmax}_{j' \in [n]}(x^\top B)_{j'}$. Let $j \in [n] \setminus (\mathcal{G} \cup \mathcal{B})$ be the index of a column satisfying the constraints from $\mathcal{H}_s^1(\alpha; l_n(s))$. Then,

$$0 \leq (x^\top B)_{j_s} - (x^\top B)_j \overset{(i)}{\leq} -x_{i_s}\frac{s}{2\alpha c\sqrt{n\log n}} + \sum_{i \in [n] \setminus \{i_s, i_s^{(1)}, \ldots, i_s^{(l_n(s))}\}} x_i(B_{i,j_s} - B_{i,j})$$

$$\leq -x_{i_s}\frac{s}{2\alpha c\sqrt{n\log n}} + \sum_{i \in [n] \setminus \{i_s, i_s^{(1)}, \ldots, i_s^{(l_n(s))}\}} x_i,$$

where in $(i)$ we used both $\mathcal{H}_s^1(\alpha; l_n(s))$ and the lower bound on $\Delta_s$ from $\mathcal{G}(\alpha)$. Next,

$$1 - x^\top A e_{j_s} \geq (1 - A_{i_s^{(l_n(s)+1)}, j_s}) \sum_{i \in [n] \setminus \{i_s, i_s^{(1)}, \ldots, i_s^{(l_n(s))}\}} x_i \overset{(i)}{\geq} \frac{s^{1/3}(l_n(s) + 1)}{2\alpha^2 cn\sqrt{n \log n}} \cdot x_{i_s}.$$

In $(i)$ we used the previous equation as well as $\mathcal{H}^2(\alpha; l_n(s))$. We recall that if $s \leq (\log n)^3$, then $l_n(s) + 1 = l_n + 1 \geq \frac{1}{8} \log n$; while if $s > (\log n)^3$ we directly have $s^{1/3}(l_n(s) + 1) \geq 2 \log n$. In all cases, we obtain

$$1 - x^\top A e_{j_s} \geq \frac{\sqrt{\log n}}{16\alpha^2 cn\sqrt{n}} \cdot x_{i_s}$$

Also, since for any $i \neq i_s$ one has $(i, j_s) \notin S$, we always have $1 - x^\top A e_{j_s} \geq \epsilon_n(1 - x_{i_s})$. In summary, under $\mathcal{E}, \mathcal{F}, \mathcal{G}(\alpha), \mathcal{H}^1(\alpha), \mathcal{H}^2(\alpha)$,

$$\forall j \in \mathcal{G}, \quad j \in \operatorname*{argmax}_{j' \in [n]}(x^\top B)_{j'} \Rightarrow 1 - x^\top A e_j \geq \min\left(\frac{c}{2}, \frac{1}{32\alpha^2 c}\right) \frac{\sqrt{\log n}}{n\sqrt{n}}. \tag{16}$$

**Edge case: $j \in \mathcal{B}$.** It only remains to focus on columns in $\mathcal{B}$. We recall that under $\mathcal{F}$, we have $|\mathcal{B}| \leq \log n$, and that compared to columns in $\mathcal{G}$, these contain two elements in $S$ instead of one. Because there are only very few of these columns, we can adapt the previous proof but with looser parameters. Conditional on $S, \mathcal{F}$, the values of $B_{i,j}$ for all $i, j \in [n]$ are i.i.d. uniform on $[0, 1]$. In particular, if we define

$$\mathcal{H}^3 := \left\{ \forall j \in \mathcal{B}, \forall i \in [n], (i, j) \in S : 1 - B_{i,j} \geq \frac{1}{\log^2 n} \right\},$$

then by the union bound we directly have

$$\mathbb{P}[(\mathcal{H}^3)^c \mid S, \mathcal{F}] \leq \frac{2|\mathcal{B}|}{\log^2 n} \leq \frac{2}{\log n}. \tag{17}$$

Next, we define

$$\mathcal{H}^4 := \left\{ \forall j \in \mathcal{B}, \exists j' \in [n] \setminus (\mathcal{G} \cup \mathcal{B}) : \forall i \in [n], (i, j) \in S : B_{i,j'} \geq \frac{1 + B_{i,j}}{2} \right\}.$$

We reason conditionally on $\mathcal{E}, \mathcal{F}, S$, and $\{B_{i,j}, (i, j) \in S\}$. By the union bound,

$$\mathbb{P}[(\mathcal{H}^4)^c \mid S, \mathcal{E}, \mathcal{F}] \leq \mathbb{P}[(\mathcal{H}^3)^c \mid S, \mathcal{F}] + \mathbb{P}[(\mathcal{H}^4)^c \mid S, \mathcal{E}, \mathcal{F}, \mathcal{H}^3]$$

$$\overset{(i)}{\leq} \frac{2}{\log n} + \sum_{j \in \mathcal{B}}\left(1 - \frac{1}{4\log^4 n}\right)^{n - |\mathcal{G} \cup \mathcal{B}|}$$

$$\overset{(ii)}{\leq} \frac{2}{\log n} + |\mathcal{B}| e^{-\frac{n}{8\log^4 n}} \leq \frac{2}{\log n} + e^{-\frac{n}{8\log^4 n}} \log n. \tag{18}$$

In $(i)$ we used the event $\mathcal{H}^3$ and in $(ii)$ we used the fact that $|\mathcal{G} \cup \mathcal{B}| \leq |S| \leq n/2$ for $n$ sufficiently large. Next, we define

$$\mathcal{H}^5 := \left\{ \forall j \in \mathcal{B}, \forall i \in [n], (i, j) \notin S : 1 - A_{i,j} \geq \frac{1}{n\log^2 n} \right\}.$$

We recall that conditional on $S, \mathcal{F}$, the values of $A_{i,j}$ for $(i, j) \notin S$ are i.i.d. stochastically dominated by a uniform on $[0, 1]$. Hence, by the union bound

$$\mathbb{P}[(\mathcal{H}^5)^c \mid S, \mathcal{F}] \leq \frac{(n - 2)|\mathcal{B}|}{n\log^2 n} \leq \frac{1}{\log n}. \tag{19}$$

Now suppose that $\mathcal{E}, \mathcal{F}, \mathcal{H}^4, \mathcal{H}^5$ hold. Fix any $j \in \mathcal{B}$ and $x \in \Delta^n$ such that $j \in \operatorname{argmax}_{j' \in [n]}(x^\top B)_{j'}$. Then, let $j' \in [n] \setminus (\mathcal{G} \cup \mathcal{B})$ corresponding to the event $\mathcal{H}^4$. For convenience, let $i_1 \neq i_2$ be the row indices for which $(i_1, j), (i_2, j) \in S$. Then, similarly as in the main

case, we have

$$0 \leq (x^\top B)_j - (x^\top B)_{j'} \overset{(i)}{\leq} -\frac{x_{i_1} + x_{i_2}}{2 \log^2 n} + \sum_{i \in [n] \setminus \{i_1, i_2\}} x_i (B_{i,j} - B_{i,j'})$$

$$\leq -\frac{x_{i_1} + x_{i_2}}{2 \log^2 n} + \sum_{i \in [n] \setminus \{i_1, i_2\}} x_i.$$

In $(i)$ we used $\mathcal{H}^3, \mathcal{H}^4$ and the definition of $j'$. On the other hand, since $\mathcal{H}^5$ holds,

$$1 - x^\top A e_j \geq \frac{1}{n \log^2 n} \sum_{i \in [n] \setminus \{i_1, i_2\}} x_i \geq \frac{x_{i_1} + x_{i_2}}{2n \log^4 n}.$$

As in the main case, we also have $1 - x^\top A e_j \geq \epsilon_n (1 - x_{i_1} + x_{i_2})$. In summary, under $\mathcal{E}, \mathcal{F}, \mathcal{H}^3, \mathcal{H}^4, \mathcal{H}^5$, we obtained

$$\forall j \in \mathcal{B}, \quad j \in \underset{j' \in [n]}{\operatorname{argmax}} (x^\top B)_{j'} \Rightarrow 1 - x^\top A e_j \geq \min \left( \frac{c}{2}, \frac{\sqrt{n}}{4 \log^{4.5} n} \right) \frac{\sqrt{\log n}}{n \sqrt{n}}. \qquad (20)$$

**Combining bounds together.** Combining Eqs. (16) and (20) together with the main decomposition from Eq. (12), we showed that under $\mathcal{E}, \mathcal{F}, \mathcal{G}(\alpha), \mathcal{H}^1(\alpha), \mathcal{H}^2(\alpha), \mathcal{H}^3, \mathcal{H}^4, \mathcal{H}^5$, we have for $n$ sufficiently large

$$1 - V(x^\star) \geq \min \left( \frac{c}{2}, \frac{1}{32 \alpha^2 c} \right) \frac{\sqrt{\log n}}{n \sqrt{n}}.$$

On the other hand, combining Eqs. (9) to (11), (13) to (15) and (17) to (19), we have

$$\mathbb{P} \left[ \mathcal{E} \cap \mathcal{F} \cap \mathcal{G}(\alpha) \cap \mathcal{H}^1(\alpha) \cap \mathcal{H}^2(\alpha) \cap \mathcal{H}^3 \cap \mathcal{H}^4 \cap \mathcal{H}^5 \right] \geq 1 - c^2 - \frac{c_8}{\sqrt{\alpha}} - \frac{c_8}{\log n},$$

for some universal constant $c_8$. As a result, we proved the desired expected bound

$$\mathbb{E} \left[ 1 - V(x^\star) \right] \geq c_9 \frac{\sqrt{\log n}}{n \sqrt{n}},$$

for some universal constant $c_9 > 0$. As a remark, by adjusting the parameters $c$ and $\alpha$, we can also get bound with high probability. $\qquad \square$

Together, Lemmas 1 and 2 prove Theorem 1.

### A.2 Proof of Theorem 2

We start by constructing a candidate solution for the defenders then lower bound its value.

**Construction of a solution to the security game.** We order the schedules by decreasing order of value for the attacker. Formally, we define

$$v_i := \max_{t \in S_i} u_a^u(t) \quad \text{and} \quad t_i := \underset{t \in S_i}{\operatorname{argmax}} \, u_a^u(t), \quad i \in [k]$$

Then, we order $[k] = \{i(1), \ldots, i(k)\}$ such that $v_{i(1)} \geq v_{i(2)} \geq \ldots \geq v_{i(k)}$. Next, we define

$$L_{\max} := \max \left\{ l \in \{1, \ldots, k\} : \sum_{s=1}^{l} \frac{v_{i(s)} - v_{i(l)}}{v_{i(s)}} < R \right\}.$$

Intuitively, by adjusting their schedules, $L_{\max}$ corresponds to the maximum index $l$ of the schedule $S_{i(l)}$ that the defenders can force the attacker to focus on. The defenders then focus on the schedule with index

$$l^* := \underset{l \in [L_{\max}]}{\operatorname{argmax}} \, u_d^u(t_{i(l)}).$$

We then compute the corresponding probabilities of coverage. We consider the following probabilities of coverage for each schedule:

$$p_i := \begin{cases} \frac{v_i - v_{i(l^*)} + \delta}{v_i}, & \text{if } i = i(l), l < l^* \\ 0 & \text{otherwise}, \end{cases}$$

where $\delta > 0$ is sufficiently small so that we still have $\sum_{i \in [k]} p_i \leq K$ and $p_i < 1$ (provided $v_{i(l^*)} > 0$). We recall that this is possible since $l^* \leq L_{\max}$. We then construct a scheduling strategy for the defenders such that each schedule $S_i$ is covered with probability $p_i$.

For instance, we can proceed as follows. We sampled a uniform distribution $U$ on $[0, 1]$. For any $r \in [R]$ let $i_r$ be the index such that

$$\sum_{i=1}^{i_r - 1} p_i < U + r \leq \sum_{i=1}^{i_r} p_i.$$

For $r = R$, there may not always be such an index, in that case we can arbitrarily choose $i_R = 1$. The final schedule is then to assign defender $r$ to $S_{i(r)}$.

**Analysis of the proposed solution.** By construction, each schedule $S_i$ has probability exactly $\min(p_i, 1)$ of being covered. Since the utilities $u_a^u(t)$ for $t \in [T]$ are sampled uniformly in $[0, 1]$, on an event $\mathcal{E}$ of probability 1 all elements $u_a^u(t)$ for $t \in [T]$ are distinct. Then, on this event we have for any $l < l^*$ that $v_{i(l)} > v_{i(l^*)}$, and hence we could always choose $\delta > 0$ such that $p_i \leq 1$.

Therefore, on $\mathcal{E}$, the expected value of target $t \in S_{i(l)}$ for $l \in [k]$ for the attacker is

$$v(t) = \begin{cases} u_a^u(t) \leq v_{i(l)} & \text{if } l \geq l^* \\ u_a^u(t)(1 - p_i) \leq v_i(1 - p_i) = v_{i(l^*)} - \delta. & \text{if } l < l^* \end{cases}$$

Further, under $\mathcal{E}$, the inequalities are strict whenever $t \neq t_{i(l)}$ and we have $v_{i(l)} < v_{i(l^*)}$ whenever $l > l^*$. In summary, the best-response for the attacker is to attack the target $t_{i(l^*)}$.

Therefore, the value of the defender under $\mathcal{E}$ is

$$V^* = u_d^u(t_{i(l^*)}) = \max_{l \in [L_{\max}]} u_d^u(t_{i(l)}),$$

where in the last equality we used the definition of $l^*$. Note that $L_{\max}$ is defined using only the utilities for the adversary $u_a^u(t)$ for $t \in [T]$. Hence, the previous derivation shows that conditionally on $L_{\max}$ and $\mathcal{E}$, $V^*$ is distributed as the maximum of $L_{\max}$ i.i.d. uniform on $[-1, 0]$ random variables, that is, as $X - 1$ where $X \sim \text{Beta}(L_{\max}, 1)$. Therefore,

$$\mathbb{E}[V^*] = \mathbb{E}[\mathbb{E}[V^* \mid L_{\max}, \mathcal{E}]\mathbb{1}[\mathcal{E}]] = -\mathbb{E}_{L_{\max}}\left[\frac{1}{L_{\max} + 1}\right], \qquad (21)$$

where we used the fact that $\mathbb{P}[\mathcal{E}] = 1$. It remains to lower bound $L_{\max}$. For any $l \in [k]$, we have

$$\sum_{s=1}^{l} \frac{v_{i(s)} - v_{i(l)}}{v_i(s)} \leq l(1 - v_{i(l)}).$$

Note that by construction, $v_1, \ldots, v_k$ are independent because they are the maximum of the utilities of the attacker on each schedule and these are disjoint. Further, $v_i \sim \text{Beta}(|S_i|, 1)$. Hence, for any $i \in [k]$ and $z \in [0, 1]$,

$$\mathbb{P}[v_i \geq 1 - z] = 1 - (1 - z)^{|S_i|} \geq 1 - e^{-z|S_i|} \geq \frac{\min(z|S_i|, 1)}{2}.$$

Now fix $l \in [k/4]$ and let $z(l) \geq 0$ such that

$$\frac{1}{2} \sum_{i \in [k]} \min(z(l)|S_i|, 1) = 2l.$$

Then, Bernstein's inequality implies that

$$\mathbb{P}[v_{i(l)} < 1 - z(l)] = \mathbb{P}\left[\sum_{i \in [k]} \mathbb{1}[v_i \geq 1 - z(l)] < l\right]$$

$$\leq \mathbb{P}\left[\sum_{i \in [k]} \mathbb{1}[v_i \geq 1 - z(l)] < \frac{1}{2} \sum_{i \in [k]} \mathbb{P}[v_i \geq 1 - z(l)]\right]$$

$$\leq \exp\left(-\frac{1}{10} \sum_{i \in [k]} \mathbb{P}[v_i \geq 1 - z(l)]\right) \leq e^{-l/5}.$$

We recall that $\alpha = \frac{\max_{i \in [k]} |S_i|}{\max_{i \in [k]} |S_i|}$ and $\sum_{i \in [k]} |S_i| = T$, which implies $|S_i| \geq \frac{T}{k\alpha}$ for all $i \in [k]$. Then, for $l < k/4$,

$$4l = \sum_{i \in [k]} \min(z(l)|S_i|, 1) \geq k \cdot \min\left(\frac{z(l)T}{\alpha k}, 1\right) = \frac{z(l)T}{\alpha},$$

where in the last inequality, we used the fact that if the minimum was 1 we would have $4l \geq k$ which contradicts the hypothesis on $l$. In summary, for any $l < k/4$, with probability at least $1 - e^{-l/5}$ we obtained

$$\sum_{s=1}^{l} \frac{v_{i(s)} - v_{i(l)}}{v_i(s)} \leq lz(l) \leq \frac{4\alpha l^2}{T}.$$

Let $\tilde{l} := \min\left(k/4, \sqrt{\frac{RT}{5\alpha}}\right)$ $l_{\max} := \lfloor \tilde{l} \rfloor$ and denote by $\mathcal{F}$ the above event for $l = l_{\max}$. Under $\mathcal{E} \cap \mathcal{F}$ we have

$$\sum_{s=1}^{l_{\max}} \frac{v_{i(s)} - v_{i(l_{\max})}}{v_i(s)} < R,$$

and as a result $L_{\max} \geq l_{\max}$. Plugging this into Eq. (21) and recalling that $\mathcal{E}$ has probability one gives

$$\mathbb{E}[V^*] \geq -\frac{1}{l_{\max} + 1} - \mathbb{P}[\mathcal{F}^c] \geq -\frac{1}{l_{\max} + 1} - e^{-l_{\max}/5}$$

$$\gtrsim -\mathbb{1}[l_{\max} = 0] - \frac{\mathbb{1}[l_{\max} \geq 1]}{l_{\max}} \gtrsim -\frac{1}{\tilde{l}} \gtrsim -\left(\frac{1}{k} + \sqrt{\frac{\alpha}{RT}}\right),$$

where the notation $\gtrsim$ only hides universal constants. This ends the proof.

## B   Algorithms and Programs

### B.1   Schedule-Finding Algorithms

Algorithm 1 and the subroutine given by Algorithm 2 are used to return valid general schedules (multiple target coverages per schedule allowed) per defender resource.

The simple schedule case is handled via a closed-form condition: for any target $t$ and home base node $h$, if the shortest path $p$ from $h$ to $t$, in a game with defense time threshold $\delta$, satisfies $\mathcal{T} \geq 2|p| + \delta - 1$, then target $t$ is considered validly defendable as a singleton schedule, as this condition implies that the game length $\mathcal{T}$ is long enough for a defender resource to travel to it from its home base, wait the required number of steps to meet the defense time threshold, and return to home base.

### B.2   Best Response Oracles

#### B.2.1   Best Response Oracles in Normal-Form Double Oracle

As part of the double oracle algorithm for solving zero-sum security games, the normal-form defender best response subroutine DEFENDERBRNF solves a mixed-integer program to determine the optimal

**Algorithm 1** Find General Schedules

---

**Require:** Graph $G = (V, E)$, start node $h$, number of timesteps $\mathcal{T}$, defense time threshold $\delta$, path cost per edge traveled $c$
**Ensure:** Set of valid schedules with associated movement costs
1: $S_{\text{simple}} \leftarrow \text{GETSIMPLESCHEDULES}(G, h, \mathcal{T}, \delta)$
2: $D \leftarrow \emptyset$                                               ▷ Initialize general schedule list
3: **for all** $t \in S_{\text{simple}}$ **do**
4:     Add $\{t\}$ to $D$ with corresponding round-trip cost
5: **end for**
6: **function** BACKTRACK($S, R$)
7:     **if** $S \neq \emptyset$ **then**
8:         $(p, \text{cost}, \text{steps}) \leftarrow \text{GETFULLPATH}(G, h, S, \delta)$
9:         **if** cost $\leq T$ **then**
10:             Add $(S, c \cdot \text{steps})$ to $D$
11:         **end if**
12:     **end if**
13:     **for** $i = 1$ to $|R|$ **do**
14:         BACKTRACK($S \cup \{R_i\}, R_{>i}$)
15:     **end for**
16: **end function**
17: BACKTRACK($\emptyset, S_{\text{simple}}$)
18: **return** $D$

---

**Algorithm 2** Get Full Path

---

1: **function** GETFULLPATH($G, h, S, \delta$)
2:     Initialize min_cost $\leftarrow \infty$, min_steps $\leftarrow \infty$, best_path $\leftarrow \emptyset$
3:     **for all** permutations $\pi$ of $S$ **do**
4:         $p \leftarrow [h]$, $c \leftarrow 0$, $s \leftarrow 0$, $u \leftarrow h$
5:         **for all** $v \in \pi$ **do**
6:             $q \leftarrow \text{SHORTESTPATH}(G, u, v)$
7:             **if** $q$ does not exist **then continue to next permutation**
8:             **end if**
9:             Append $q \setminus \{u\}$ to $p$                          ▷ Avoid duplicate nodes
10:             $c \leftarrow c + |q| - 1$, $s \leftarrow s + |q| - 1$
11:             Append $v$ to $p$ ($\delta - 1$) times                          ▷ Dwell at $v$
12:             $c \leftarrow c + \delta - 1$
13:             $u \leftarrow v$
14:         **end for**
15:         $r \leftarrow \text{SHORTESTPATH}(G, u, h)$
16:         **if** $r$ does not exist **then continue to next permutation**
17:         **end if**
18:         Append $r \setminus \{u\}$ to $p$
19:         $c \leftarrow c + |r| - 1$, $s \leftarrow s + |r| - 1$
20:         **if** $c < $ min_cost **then**
21:             best_path $\leftarrow p$, min_cost $\leftarrow c$, min_steps $\leftarrow s$
22:         **end if**
23:     **end for**
24:     **if** best_path $= \emptyset$ **then**
25:         **return** $(\text{None}, \infty, \infty)$
26:     **else**
27:         **return** (best_path, min_cost, min_steps)
28:     **end if**
29: **end function**

---

joint strategy for the defender, given a fixed distribution over attacker actions. This best response accounts for spatial movement constraints, home base assignments, and a defense time threshold $\delta$.

Let $G = (V, E)$ be a directed graph over which defender movement is defined, and let $\mathcal{T}$ denote the number of timesteps in the game. The defender team consists of $D$ mobile resources, each with a valid home base set $H_d \subseteq V$. Let $T \subseteq V$ be the full set of targets, and let $T_a \subseteq T$ denote the subset of targets currently selected by the attacker. Each $t \in T_a$ is associated with a target value $V_t$ and a probability of attack $P_t$.

We define binary variables $v_{i,\tau}^{(d)} \in \{0, 1\}$ to indicate whether defender $d$ is at node $i$ at timestep $\tau \in \{0, \dots, \mathcal{T}\}$, and $g_t \in \{0, 1\}$ to indicate whether target $t \in T_a$ is successfully interdicted. The DEFENDERBRNF optimization is given by:

$$
\begin{aligned}
\max_{v,g} \quad & \sum_{t \in T_a} P_t \cdot V_t \cdot g_t \\
\text{s.t.} \quad & \sum_{i \in H_d} v_{i,0}^{(d)} = 1, \quad \sum_{i \in H_d} v_{i,\mathcal{T}}^{(d)} = 1 \quad \forall d \in \{1, \dots, D\} && \text{(Start and end at home base)} \\
& v_{i,0}^{(d)} = 0, \quad v_{i,\mathcal{T}}^{(d)} = 0, \quad \forall i \notin H_d, \ \forall d && \text{(No start/end outside home base)} \\
& v_{i,\tau}^{(d)} \leq \sum_{j \in \mathcal{N}(i)} v_{j,\tau-1}^{(d)}, \quad \forall i \in V, \ \forall \tau \in \{1, \dots, \mathcal{T}\}, \ \forall d && \text{(Feasible movements)} \\
& \sum_{i \in V} v_{i,\tau}^{(d)} = 1, \quad \forall \tau \in \{0, \dots, \mathcal{T}\}, \ \forall d && \text{(Single location per timestep)} \\
& \delta \cdot g_t \leq \sum_{\tau=0}^{\mathcal{T}-1} \sum_{d=1}^{D} v_{t,\tau}^{(d)}, \quad \forall t \in T_a && \text{(Interdiction threshold)} \\
& v_{i,\tau}^{(d)} \in \{0, 1\}, \quad g_t \in \{0, 1\}
\end{aligned}
$$

Here, $\mathcal{N}(i)$ denotes the in-neighbors of node $i$ (including self-loops for waiting). Constraints enforce defender path feasibility, ensure one location per timestep, and require sufficient visitation to interdict each target in $T_a$.

For our Normal-Form Double Oracle algorithm, we define the ATTACKERBRNF as selecting the $k$ targets with highest expected attacker value, where utility is discounted by the probability $q_t$ that target $t$ is interdicted under the current defender strategy $x_d$. Formally, the attacker solves:

$$
\text{ATTACKERBRNF}(x_d) = \underset{S \subseteq T, \, |S| = k}{\arg\max} \sum_{t \in S} V_t \cdot (1 - q_t)
$$

where $T$ is the set of all targets, $V_t$ is the value for target $t$, and $q_t$ is the interdiction probability of $t$ induced by defender strategy $x_d$. The attacker selects the top-$k$ targets with the highest discounted expected utility.

### B.2.2 Best Response Oracles in Schedule-Form Double Oracle

In schedule-form security games, each defender resource selects a schedule—a subset of targets it can reach and cover—subject to mobility and defense time threshold constraints. Given a fixed mixed strategy of the attacker, the defender's best response selects one schedule per resource to maximize expected utility. Let $\mathcal{R}$ denote the set of defender resources, $T$ the set of targets, and $\mathcal{S}_r \subseteq 2^T$ the set of feasible schedules for defender resource $r \in \mathcal{R}$. Let $w_t$ denote the expected utility of defending target $t$, computed from the current attacker strategy.

We define binary variables $x_{r,i} \in \{0, 1\}$ indicating selection of the $i^{\text{th}}$ schedule in $\mathcal{S}_r$, and $g_t \in \{0, 1\}$ indicating whether target $t$ is covered. The defender best response DEFENDERBRSF solves:

$$\min_{x,g} \quad -\sum_{t \in T} w_t \cdot g_t$$

$$\text{s.t.} \quad \sum_{i=1}^{|\mathcal{S}_r|} x_{r,i} = 1, \quad \forall r \in \mathcal{R} \qquad \qquad \text{(One schedule per resource)}$$

$$g_t \le \sum_{r \in \mathcal{R}} \sum_{i:t \in \mathcal{S}_r[i]} x_{r,i}, \quad \forall t \in T \qquad \text{(Coverage implies schedule selection)}$$

$$x_{r,i} \in \{0,1\}, \quad g_t \in \{0,1\}$$

To compute the attacker best response, we assume the attacker selects the single target $t \in T$ with the highest expected utility under the defender's current mixed strategy. Let $D_d(j)$ be the probability assigned to defender pure strategy $j$, and let $t \in S_j^\tau$ denote coverage of target $t$ at timestep $\tau$ under schedule $j$. The attacker solves:

$$\text{ATTACKERBRSF}(x_d) = \arg\max_{t \in T} \sum_j D_d(j) \cdot \begin{cases} V_{c,t}^a & \text{if } t \in S_j \\ V_{uc,t}^a & \text{otherwise} \end{cases}$$

The attacker computes expected utility using the coverage status of each target across all defender schedules, receiving $V_{c,t}^a$ if $t$ is ever covered, and $V_{uc,t}^a$ otherwise.

All experiments using best response oracles were computed with equilibrium gap tolerance $\epsilon = 10^{-12}$.

### B.3 Time Constrained Depth First Search

To enumerate the full set of valid movement actions for a team of mobile units on a graph, we employ a time-constrained depth-first search (DFS). This recursive procedure explores all paths of a fixed length $\mathcal{T}$, starting from allowed initial nodes and optionally constrained by required end nodes or a return-to-start condition. Waiting behavior is supported by including self-loops when allowed. For multiple units, individual path sets are generated and combined via Cartesian product, then reformatted into time-expanded action matrices for use in normal-form action space construction.

---

**Algorithm 3** GenerateMovingPlayerActions$(G, m, \mathcal{S}, \mathcal{T}, \texttt{wait}, \mathcal{E}, \texttt{return})$

---

**Require:** Graph $G = (V, E)$, number of units $m$, start node sets $\mathcal{S} = (S_1, \ldots, S_m)$, number of timesteps $\mathcal{T}$, waiting flag $\texttt{wait}$, end node sets $\mathcal{E} = (E_1, \ldots, E_m)$ (optional), return-to-start flag $\texttt{return}$

**Ensure:** Set of valid movement actions for $m$ units over $\mathcal{T}$ timesteps
1: **if** $m = 0$ **then**
2:      **return** $\emptyset$
3: **end if**
4: **if** $\mathcal{S}$ or $\mathcal{E}$ are not length $m$ **then**
5:      **raise error**
6: **end if**
7: **if** $\mathcal{S}$ is empty **then**
8:      $\mathcal{S} \leftarrow (V, \ldots, V)$
9: **end if**
10: **if** $\mathcal{E}$ is empty **then**
11:      $\mathcal{E} \leftarrow (V, \ldots, V)$
12: **end if**
13: **for** $i = 1$ **to** $m$ **do**
14:      $\mathcal{P}_i \leftarrow$ GeneratePaths$(G, S_i, E_i, \mathcal{T}, \texttt{wait}, \texttt{return})$
15: **end for**
16: $\mathcal{A} \leftarrow$ all Cartesian products of $(\mathcal{P}_1, \ldots, \mathcal{P}_m)$
17: Format each joint action in $\mathcal{A}$ into timestep-major form
18: **return** $\mathcal{A}$

---

**Algorithm 4** GeneratePaths($G, S, E, \mathcal{T}, \texttt{wait}, \texttt{return}$)

---

**Require:** Graph $G = (V, E)$, valid start nodes $S$, end nodes $E$, timesteps $\mathcal{T}$, waiting flag $\texttt{wait}$, return-to-start flag $\texttt{return}$
**Ensure:** All valid paths of length $\mathcal{T}$ from $S$ to $E$
 1: Initialize $\texttt{all\_paths} \leftarrow \emptyset$
 2: **for all** $s \in S$ **do**
 3:     DFS($[s], s$)
 4: **end for**
 5: **return** $\texttt{all\_paths}$
 6: **function** DFS($\texttt{path}, \texttt{origin}$)
 7:     **if** $|\texttt{path}| = \mathcal{T}$ **then**
 8:         **if** $\texttt{path}[-1] \in E$ **or** $\texttt{return}$ **and** $\texttt{path}[-1] = \texttt{origin}$ **then**
 9:             Add $\texttt{path}$ to $\texttt{all\_paths}$
10:         **end if**
11:         **return**
12:     **end if**
13:     $v \leftarrow \texttt{path}[-1]$
14:     $\mathcal{N} \leftarrow \texttt{neighbors}(v)$
15:     **if** $\texttt{wait}$ or $\mathcal{N} = \emptyset$ **then**
16:         DFS($\texttt{path} + [v], \texttt{origin}$)
17:     **end if**
18:     **for all** $u \in \mathcal{N}$ **do**
19:         DFS($\texttt{path} + [u], \texttt{origin}$)
20:     **end for**
21: **end function**

---

## C  Framework Details

This Appendix details the the the GUARD game class hierarchy in greater detail than the main text. For exact implementation details, please visit our public Git repository for this project at `https://github.com/CoffeeAndConvexity/GUARD`.

### C.1  Graph Game Parameters

The *Graph Game* class serves as the foundational layer of the GUARD architecture and is responsible for configuring dynamic multi-agent game environments on graphs. It supports customizable parameters to model general-sum and zero-sum security games with both moving and stationary resources.

**Core Parameters.** The game is initialized with the following attributes:

- A directed graph $G = (V, E)$ representing the environment.
- A time horizon $\mathcal{T}$ specifying the number of timesteps.
- Resource counts for each player: number of moving/stationary attackers and defenders.
- Allowed start and end node sets for moving players (e.g., patrol bases or entry points).
- Interdiction protocol, which determines how target interdiction occurs, including time threshold and capture radius parameters.
- A set of `Target` objects, each with attacker and defender value attributes.
- Additional game constraints: waiting permissions (`allow_wait`) and forced return to origin for moving resources (`force_return`).

**Action Space Generation.** The `generate_actions(player)` method constructs a valid joint action space for either the attacker or defender, handling both moving and stationary resources. Moving resource actions are computed using a constrained depth-first search that respects home bases, end nodes, waiting rules, and return constraints (see Appendix B.3). Stationary attacker

actions select from subsets of the target nodes (or include `None`), while stationary defender actions are generated via Cartesian products of valid placements. Combined action paths are returned in timestep-major form as $\mathcal{T} \times N$ arrays.

**Utility Evaluation.** The `evaluate_actions(defender_action, attacker_action)` method simulates interactions between defender and attacker paths using the game's interdiction protocol. Attacker utilities are calculated by summing the values of targets reached without being interdicted. Defender values are represented as the negation of attacker scores in zero-sum settings or separately in general-sum settings. Interdiction for moving attackers is radius-based and computed at each timestep; interdiction for stationary attackers is governed by a defense time threshold requiring $\delta$ timesteps spent by defender resources at attacker-selected targets.

**Utility Matrix Construction.** The `generate_utility_matrix(general_sum, defender_step_cost)` method constructs the full game matrix by evaluating all defender-attacker action pairs. In zero-sum mode, the matrix $U \in \mathbb{R}^{|\mathcal{A}_d| \times |\mathcal{A}_a|}$ is normalized by its maximum absolute value. In general-sum mode, two separate matrices are returned for attacker and defender utilities, with optional penalties applied for defender movement steps.

**General-Sum Evaluation.** The `evaluate_actions_general` method extends utility computation by incorporating defender path costs. This enables richer modeling of path efficiency and asymmetric incentives.

### C.2 Security Game Parameters

The *Security Game* class extends the general *Graph Game* layer to implement a canonical Stackelberg Security Game abstraction. It enforces a fixed role structure with **stationary attackers** (who each select a single target) and **moving defenders** (who patrol over a graph from designated home bases). This class is designed for both zero-sum and general-sum formulations and supports both normal-form and schedule-form representations.

**Core Assumptions.** The class initializes with:

- Only **stationary attackers**: no attacker movement paths are generated.
- Only **moving defenders**: all defender units are assigned to start and end nodes (typically enforcing home base return).
- An `InterdictionProtocol` object, which governs coverage rules based on defense time thresholds and graph-based distances.

**Schedule Form Capabilities.** In schedule-form mode, each defender is assigned a set of feasible *schedules* (i.e., subsets of targets reachable within the time horizon $\mathcal{T}$ while satisfying interdiction dwell time requirements and optional return constraints). These schedules are generated via a recursive backtracking procedure (see Algorithm 1) and then deduplicated.

The class supports:

- Defender schedule enumeration: `find_valid_schedules(start_node)`.
- Cost-aware joint schedule assembly: `generate_defender_actions_with_costs`.
- Utility matrix generation: `generate_schedule_game_matrix`, with general-sum support and normalization.
- Target utility assignment using attacker and defender multipliers, optionally randomized for experimental baselines.

**Target Utility Matrix.** Each target is assigned a utility tuple reflecting covered/uncovered values for each team. These are stored as a $4 \times |T|$ matrix:

- Row 0: Defender utility if uncovered
- Row 1: Defender utility if covered

- Row 2: Attacker utility if covered
- Row 3: Attacker utility if uncovered

This matrix can be optionally scaled or randomized to reflect asymmetric preferences or experimental perturbations.

**Output.** The final dictionary returned from `schedule_form(...)` includes defender schedules, actions, and (if enabled) attacker and defender utility matrices. These outputs are formatted for compatibility with double oracle solvers, matrix-based Stackelberg optimization, or heuristic evaluations.

This abstraction isolates core Stackelberg security-game functionality while remaining modular enough to support domain-specific modeling in higher layers, including target value scaling, defender mobility constraints, and payoff asymmetries.

### C.3 Domain-Specific Game Parameters

The *GreenSecurityGame* and *InfraSecurityGame* classes extend the *SecurityGame* layer to support input data processing and target construction. Both game types ultimately instantiate and return a configured `SecurityGame` object, but differ in how graphs and targets are derived.

**Common Parameters.** Both GSG and ISG classes accept the following arguments during generation:

- `num_attackers`: Number of stationary attackers (e.g., poachers or saboteurs).
- `num_defenders`: Number of mobile defender units.
- `home_base_assignments`: List of coordinate locations (lat, lon) from which each defender unit may begin/end patrol.
- `num_timesteps`: Total number of time steps in the game.
- `interdiction_protocol`: Interdiction rules (e.g., defense time threshold).
- `defense_time_threshold`: Number of defender steps at a target required for successful interdiction.
- `generate_utility_matrix`: Boolean flag to compute the full normal-form utility matrix.
- `generate_actions`: Whether to enumerate full defender and attacker action sets.
- `force_return`: If true, all defender paths must return to their starting location.
- `schedule_form`: If true, schedule-form representation is used.
- `general_sum`: Enables general-sum game formulation with asymmetric target values.
- `alpha`: Controls strength of escape-proximity influence on attacker utility.
- `random_target_values`: Overwrites target values with random values if true (zero-sum only).
- `randomize_target_utility_matrix`: Randomizes utility entries in the target matrix (for experiments).

**Green Security Games.** As detailed in the main text, GSGs define their graph using a custom spatial grid over a bounding box. Each cell contains animal tracking data and is scored using either `density` (raw frequency of observations) or `centroid` (K-means clustering with cluster weights). Optional escape-line proximity adjustments are applied to model attacker incentives for exit routes. Targets are created using the cell center, population density, and animal value scaling factors.

Additional parameters specific to GSGs include:

- `scoring_method`: "centroid" or "density", for scoring animal activity.
- `num_clusters`: Number of centroid clusters (for centroid mode).
- `num_rows`, `num_columns`: Grid resolution.

- `escape_line_points`: Two-point line representing an attacker escape boundary for computing distance-to-escape metrics.
- `attacker_animal_value`: Scaling factor for attacker utilities based on species value (functions to normalize payoffs).
- `defender_animal_value`: Scaling factor for defender valuation (e.g., tourism, biodiversity - functions to normalize payoffs).

**Infrastructure Security Games.** As mentioned in the main text, ISGs use OpenStreetMap street networks and power infrastructure feature data to construct real-world graphs. Nodes are street intersections, and targets are mapped from power infrastructure features to the closest node. Target values are computed using population estimates (from Census block data) and infrastructure type multipliers.

Additional parameters specific to ISGs include:

- `infra_df`: DataFrame of infrastructure features with coordinates and types.
- `block_gdf`: Census block GeoDataFrame with geometry and population counts.
- `infra_weights`: Dictionary of weights per infrastructure type (e.g., medical clinic, power_tower).
- `mode`: Either ''block'' or ''radius'' to determine population assignment method.
- `escape_point`: Coordinate (x, y) used to assign escape proximity-based scaling.
- `population_scaler`: Exponent applied to log-population terms in score computation.
- `attacker_feature_value`, `defender_feature_value`: Scaling weights for general-sum target utilities, serves to normalize payoffs.

**General-Sum and Schedule-Form Options.** If general-sum mode is enabled, the system allows asymmetric target coverage scaling factors:

- `attacker_penalty_factor`: Scales down (positive) gained attacker value when a target is covered by a defender schedule.
- `defender_penalty_factor`: Scales down (negative) incurred defender value when a target is covered by a defender schedule.
- `defender_step_cost`: Optional cost incurred by defenders per movement step.

If schedule-form mode is enabled:

- `simple`: If True, only single-target schedules are considered.
- The system generates valid schedules per defender and computes either full utility matrices or reduced representations as needed.

**Output.** Each domain-specific game returns a fully parameterized `SecurityGame` instance, with optional attributes: `defender_actions`, `attacker_actions`, utility matrices (`utility_matrix`, `attacker_utility_matrix`, `defender_utility_matrix`), and schedule-form dictionaries, depending on input configuration.

### C.4 Overpass Turbo Query for OSM NYC Civil Infrastructure

```
[out:json][timeout:25];
(
  // Healthcare
  node["amenity"="hospital"](40.4774, -74.2591, 40.9176, -73.7004);
  way["amenity"="hospital"](40.4774, -74.2591, 40.9176, -73.7004);
  relation["amenity"="hospital"](40.4774, -74.2591, 40.9176, -73.7004);
```

```
   node["amenity"="clinic"](40.4774, -74.2591, 40.9176, -73.7004);
   way["amenity"="clinic"](40.4774, -74.2591, 40.9176, -73.7004);
   relation["amenity"="clinic"](40.4774, -74.2591, 40.9176, -73.7004);

   // Education
   node["amenity"="school"](40.4774, -74.2591, 40.9176, -73.7004);
   way["amenity"="school"](40.4774, -74.2591, 40.9176, -73.7004);
   relation["amenity"="school"](40.4774, -74.2591, 40.9176, -73.7004);

   node["amenity"="university"](40.4774, -74.2591, 40.9176, -73.7004);
   way["amenity"="university"](40.4774, -74.2591, 40.9176, -73.7004);
   relation["amenity"="university"](40.4774, -74.2591, 40.9176, -73.7004);

   // Water & Sanitation
   node["man_made"="water_works"](40.4774, -74.2591, 40.9176, -73.7004);
   way["man_made"="water_works"](40.4774, -74.2591, 40.9176, -73.7004);
   relation["man_made"="water_works"](40.4774, -74.2591, 40.9176, -73.7004);

   node["man_made"="wastewater_plant"](40.4774, -74.2591, 40.9176, -73.7004);
   way["man_made"="wastewater_plant"](40.4774, -74.2591, 40.9176, -73.7004);
   relation["man_made"="wastewater_plant"](40.4774, -74.2591, 40.9176, -73.7004);

   // Energy Utilities
   node["power"](40.4774, -74.2591, 40.9176, -73.7004);
   way["power"](40.4774, -74.2591, 40.9176, -73.7004);
   relation["power"](40.4774, -74.2591, 40.9176, -73.7004);

   // Government and Emergency Services
   node["amenity"="fire_station"](40.4774, -74.2591, 40.9176, -73.7004);
   way["amenity"="fire_station"](40.4774, -74.2591, 40.9176, -73.7004);
   relation["amenity"="fire_station"](40.4774, -74.2591, 40.9176, -73.7004);

   node["amenity"="police"](40.4774, -74.2591, 40.9176, -73.7004);
   way["amenity"="police"](40.4774, -74.2591, 40.9176, -73.7004);
   relation["amenity"="police"](40.4774, -74.2591, 40.9176, -73.7004);

   node["amenity"="courthouse"](40.4774, -74.2591, 40.9176, -73.7004);
   way["amenity"="courthouse"](40.4774, -74.2591, 40.9176, -73.7004);
   relation["amenity"="courthouse"](40.4774, -74.2591, 40.9176, -73.7004);

   // Critical Infrastructure
   node["man_made"="bunker_silo"](40.4774, -74.2591, 40.9176, -73.7004);
   way["man_made"="bunker_silo"](40.4774, -74.2591, 40.9176, -73.7004);
   relation["man_made"="bunker_silo"](40.4774, -74.2591, 40.9176, -73.7004);

   // Communications
   node["man_made"="communications_tower"](40.4774, -74.2591, 40.9176, -73.7004);
   way["man_made"="communications_tower"](40.4774, -74.2591, 40.9176, -73.7004);
   relation["man_made"="communications_tower"](40.4774, -74.2591, 40.9176, -73.7004);
);
out body;
>;
out skel qt;
```

## C.5    Data Licenses and Attribution

All data used for building games with GUARD (movebank, OSM, Census) do not include personally
identifiable information or offensive content. All elephant study data is publicly available from the

Movebank website [49] and with creative commons (CC) [32] and CC BY-NC 4.0 [22] licenses permitting use without direct consent with correct attribution and non-commercial applications.

## C.6   Limitations

The GUARD framework exhibits several scalability limitations when applied to dense environments or large parameter configurations. In particular, for path-based NFG formulations over highly connected graphs, generating instances with a large number of timesteps (e.g., 11 or more) while requesting explicit computation of action sets and utility matrices can lead to prohibitively slow game construction. At this scale, the number of valid defender paths can exceed $10^6$, rendering the corresponding utility matrices too large for most computation resources. Introducing multiple defender resources further exacerbates this issue, as the defender's action space grows exponentially with the number of defenders due to the combinatorial nature of path assignments across resources, as illustrated in Figure 1.

While SFG formulations are less sensitive to the number of timesteps in terms of action space growth (actions are defined over defender schedules rather than full paths) the general schedule construction algorithm described in Section B.1 also becomes computationally burdensome under large timesteps or when the number of targets is high. This is due to the need to enumerate all feasible combinations of targets and compute valid patrol tours within the time horizon. Then, once generated, large utility matrices in both NFG and SFG formats can lead to memory limitations or significant slowdowns when passed to game-theoretic solvers that rely on linear programming. Notably, we encountered such issues when attempting to run sparsity experiments on the larger-scale Chinatown ISG instances; resolving these would likely require machines with greater memory capacity or improved parallelization support, especially when using support-bounded MIP solvers.

Additionally, the current definitions of targets and actions can result in degenerate game structures. For instance, if a game instance includes targets that are unreachable by defenders due to resource constraints, home base restrictions, or short time horizons, attackers will trivially exploit those undefended targets in best-response strategies. Such cases are feasible in real-world domains and highlight the need for enhanced constraint modeling.

Finally, extending the framework to support richer real-world constraints—such as assigning defenders to restricted subgraphs or modeling role-specific access—would further improve the realism and applicability of GUARD to practical security domains.

## C.7   Potential Negative Societal Impacts

While our framework is designed to aid defenders in securing vulnerable assets using realistic data-driven security game instances, it could in principle be misused to simulate or optimize attacker behavior. For instance, adversaries could leverage the framework's capabilities to test and refine evasion strategies against known defensive patrol patterns in infrastructure or poaching contexts. Additionally, access to detailed domain-specific game instances derived from real-world data could reveal sensitive information and security details for real-world environments.

To mitigate such risks, we introduce several precautions. First, researchers and practitioners using the framework should refrain from publicly releasing sensitive configuration files, particularly those derived from real defense operations or infrastructure schematics (all data we utilize is publicly available and does not compromise any specific assets). Second, future development can incorporate access controls for datasets and models flagged as high-risk. Finally, if usage/development becomes widespread, collaboration with domain experts can ensure responsible deployment and use of the GUARD framework, particularly in settings involving law enforcement, conservation, or urban infrastructure.

## C.8   Pre-Defined Game Instance Details

The following are tables outlining parameter specifications of the pre-generated zero sum game instances currently available via import in the GUARD library or at `https://github.com/CoffeeAndConvexity/GUARD` in the /predefined_games directory as .pkl files.

**Lobéké National Park GSG Instances - Preprocessing shown in Appendix D.1**

| Instance | Type | C | Grid | $\mathcal{T}$ | A | D | $\delta$ | FR | S | Size |
|---|---|---|---|---|---|---|---|---|---|---|
| lobeke_nfg_gsg_1.pkl | NFG | 12 | $7 \times 7$ | 7 | 1 | 1 | 1 | F | 7 | $9{,}075 \times 12$ |
| lobeke_nfg_gsg_2.pkl | NFG | 10 | $7 \times 7$ | 8 | 2 | 1 | 1 | F | 8 | $41{,}479 \times 55$ |
| lobeke_sfg_gsg_1.pkl | SFG | 10 | $7 \times 7$ | 7 | 1 | 3 | 2 | T | 9 | $24{,}389 \times 10$ |
| lobeke_sfg_gsg_2.pkl | SFG | 12 | $8 \times 8$ | 9 | 1 | 2 | 2 | T | 9 | $1{,}849 \times 12$ |

Table 2: Pre-generated GSG instances for Lobéké National Park. C = Num. Elephant Cluster Targets, Grid = Game gridworld dimensions $\mathcal{T}$ = Timesteps, A = Attackers, D = Defenders, $\delta$ = Defense Time Threshold, FR = Force Return, S = Nash Support, Size = Utility matrix dimensions ($|A_d| \times |A_a|$), T = True, F = False.

**Etosha National Park GSG Instances**

The Etosha elephant dataset contains 1,275,235 observations of 8 unique elephants from 2008 to 2012 [22].

The Etosha National Park bounding box used was:

- `lat_min` $= -19.41637$
- `lat_max` $= -19.06224$
- `lon_min` $= 16.22564$
- `lon_max` $= 16.83427$

Home bases for the Etosha National Park GSG were:

- Halali camp and tourist area at (-19.03683, 16.47170)
- Collection of safari/conservation attraction sites = (-19.31668, 16.87791)
- Park outpost on Ondongab road = (-19.20540, 16.19422)

Preprocessing for the Etosha dataset (nearly identical to the Lobéké preprocessing in Appendix D.1, and unused in our experiments) can be found in the GUARD Git repo.

| Instance | Type | C | Grid | $\mathcal{T}$ | A | D | $\delta$ | FR | S | Size |
|---|---|---|---|---|---|---|---|---|---|---|
| etosha_nfg_gsg_1.pkl | NFG | 8 | $6 \times 6$ | 9 | 1 | 1 | 2 | F | 5 | $32{,}367 \times 9$ |
| etosha_nfg_gsg_2.pkl | NFG | 8 | $6 \times 6$ | 9 | 2 | 1 | 1 | T | 6 | $23{,}323 \times 36$ |
| etosha_sfg_gsg_1.pkl | SFG | 7 | $6 \times 6$ | 8 | 1 | 2 | 2 | T | 7 | $169 \times 7$ |
| etosha_sfg_gsg_2.pkl | SFG | 7 | $8 \times 8$ | 10 | 1 | 2 | 1 | T | 6 | $121 \times 7$ |

Table 3: Pre-generated GSG instances for Etosha National Park.

**Chinatown ISG Instances - Preprocessing shown in Appendix D.1**

| Instance | Type | Region | $\mathcal{T}$ | A | D | $\delta$ | FR | S | Size |
|---|---|---|---|---|---|---|---|---|---|
| chinatown_nfg_isg_1.pkl | NFG | Full | 6 | 2 | 1 | 2 | T | 5 | $3{,}852 \times 276$ |
| ne_chinatown_sfg_isg_1.pkl | SFG | NE | 7 | 1 | 1 | 1 | T | 8 | $50 \times 13$ |
| ne_chinatown_sfg_isg_2.pkl | SFG | NE | 8 | 1 | 1 | 1 | T | 8 | $76 \times 13$ |
| nw_chinatown_sfg_isg_1.pkl | SFG | NW | 8 | 1 | 1 | 1 | T | 4 | $23 \times 8$ |

Table 4: Pre-generated ISG instances for Chinatown area of NYC.

## C.9 Differentiation from Gamut

While GAMUT does generate games with structured types (e.g., coordination, zero-sum), its utility assignments remain randomized and detached from any real-world dynamics. In contrast, GUARD uses open-access data sources to define not only the game structure but also the utility values,

enabling a more realistically-grounded game environment. Additionally, the types of games offered by GAMUT differ substantially from those in GUARD, with GAMUT's graph games lacking the capacity to incorporate real-world data into target values or network structure. Overall, we do not consider Gamut a direct competitor due to differences in objective and methodology.

# D  Additional Experiment Information and Results

In this section we provide extra details on how experiments were carried out including code snippets for preprocessing and game generation, additional parameter settings, and metric values we set to generate the games for the experiments. We also include some extra experiment results that were not reported in the main text. **Note:** The number of timesteps reported in this appendix and throughout the paper differs from the parameter `num_timesteps` in the code, as this variable actually refers to the number of game states, which includes the zeroth state, so `num_timesteps` $= \mathcal{T}$ in a code snippet actually refers to a $\mathcal{T} - 1$ timestep game.

### Compute Resources

Experiments were conducted on a Windows 64-bit machine with Intel(R) Core(TM) i9-14900KF, 3.20 GHz, with access to 64GB of RAM. All three classes of experiments maxed out this machine's RAM in the larger parameter settings, so all experiments required the total 64GB. We used Gurobi 10.0.3 (Gurobi Optimization, LLC 2023) for (MI)LPs.

## D.1  Data Preprocessing and Sample Game Generation

This section provides reference code used to preprocess data and instantiate sample Green and Infrastructure Security Game objects. Each domain requires cleaning real-world input data and converting it into the format required by the domain-specific security game constructor. The following code blocks also contain coordinate bounding boxes used to filter geospatial data to the correct location for experiments.

**Green Security Game.**  GSG instances are constructed from elephant GPS collar data collected in Lobéké National Park. The raw data includes multiple collar devices across different elephants and time periods. The following Python code concatenates and filters the raw datasets, extracting only relevant fields:

Listing 1: Preprocessing raw elephant collar datasets into GSG input.

```python
import pandas as pd

# Assume 'datasets' is a list of 6 individual CSVs from Movebank
datasets = [
    pd.read_csv("collar_14118.csv"),
    pd.read_csv("collar_14120.csv"),
    pd.read_csv("collar_39839.csv"),
    pd.read_csv("collar_46179.csv"),
    pd.read_csv("collar_39840.csv"),
    pd.read_csv("collar_47574.csv"),
]

# Columns of interest for location and metadata
columns_to_keep = [
    "event-id", "timestamp", "location-long", "location-lat",
    "individual-local-identifier"
]

# Clean and unify all datasets
cleaned = []
for df in datasets:
    df = df[columns_to_keep].copy()
    df.columns = ["id", "timestamp", "lon", "lat", "animal_id"]
    cleaned.append(df.dropna())
```

```
# Concatenate and write to final CSV
lobeke_df = pd.concat(cleaned, ignore_index=True)
# Save optional: lobeke_df.to_csv("lobeke.csv", index=False)
```

After preprocessing, the final dataframe `lobeke_df` is passed into the game generator. Below is the code for instantiating a GSG instance with centroid-based scoring:

Listing 2: Instantiating a sample Green Security Game.

```python
from security_game.green_security_game import GreenSecurityGame

# Define Lobeke National Park bounding box
coordinate_rectangle = [2.0530, 2.2837, 15.8790, 16.2038]

# Define defender bases and escape line
boulou_camp = (2.2, 15.9)
kabo_djembe = (2.0532, 16.0857)
bomassa = (2.2037, 16.1870)
inner_post = (2.2, 15.98)
sangha_river = [(2.2837, 16.1628), (2.053, 16.0662)]

schedule_form_kwargs = {
    "schedule_form": True,
    "simple": True,
    "attacker_penalty_factor": 5,
    "defender_penalty_factor": 5,
}
general_sum_kwargs = {
    "general_sum": True,
    "attacker_animal_value": 2350,
    "defender_animal_value": 22966,
    "defender_step_cost": 0,
}

# Create and generate game instance
gsg = GreenSecurityGame(
    lobeke_df, coordinate_rectangle, "centroid",
    num_clusters=10, num_rows=7, num_columns=7,
    escape_line_points=sangha_river
)

gsg.generate(
    num_attackers=1,
    num_defenders=2,
    home_base_assignments=[(kabo_djembe, bomassa, inner_post) for _ in
        range(2)],
    num_timesteps=8,
    defense_time_threshold=1,
    force_return=True,
    generate_utility_matrix=False,
    **schedule_form_kwargs,
    **general_sum_kwargs
)
```

**Infrastructure Security Game.** The ISG is built from a geospatial dataset of power, healthcare, water, and communication infrastructure in Manhattan's Chinatown. The following code preprocesses and flattens infrastructure entries into point representations with standardized types:

Listing 3: Preprocessing OSM-derived infrastructure data.

```python
import geopandas as gpd
import pandas as pd
```

```python
gdf = gpd.read_file("chinatown_infra.geojson")

# Standardize and extract infrastructure types
infra_columns = ["id", "name", "power", "man_made", "amenity", "
    generator:method", "generator:source", "geometry"]
gdf = gdf[[col for col in infra_columns if col in gdf.columns]].copy()

# Construct unified 'type' column from available tags
gdf["type"] = gdf["power"].combine_first(gdf["amenity"]).combine_first
    (gdf["man_made"])

# Flatten nodes and ways into point data
df_nodes = gdf[gdf["id"].str.contains("node")].copy()
df_nodes["x"] = df_nodes.geometry.x
df_nodes["y"] = df_nodes.geometry.y
df_nodes = df_nodes.drop(columns=["geometry"])

df_ways = gdf[gdf["id"].str.contains("way")].copy()
df_ways = df_ways.to_crs("EPSG:32618")
df_ways["centroid"] = df_ways.geometry.centroid
df_ways = df_ways.set_geometry("centroid").to_crs("EPSG:4326")
df_ways["x"] = df_ways.geometry.x
df_ways["y"] = df_ways.geometry.y
df_ways = df_ways.drop(columns=["geometry", "centroid"])

# Combine into one unified DataFrame
df_combined = pd.concat([df_nodes, df_ways], ignore_index=True)[["id",
    "name", "type", "x", "y"]]
```

Listing 4: Instantiating a sample Infrastructure Security Game.

```python
from security_game.infra_security_game import InfraSecurityGame

ny_blocks_gdf = gpd.read_file("tl_2020_36_tabblock20.shp")

# User-specified set of relative feature importances
INFRA_WEIGHTS = {
    "plant": 1.5, "solar_generator": 0.95, "hospital": 1.5,
    "school": 1.25, "water_works": 1.45, "fire_station": 1.3,
    "communications_tower": 1.25, "pole": 0.85, "tower": 1.1,
    # truncated for brevity
}

bbox_downtown_large = (40.7215, 40.710, -73.9935, -74.010)
fifth_precinct = (40.7163, -73.9974)
booking_station = (40.7162, -74.0010)
police_plaza = (40.7124, -74.0017)
troop_nyc = (40.7166, -74.0064)
first_precinct = (40.7204, -74.0070)

schedule_form_kwargs = {
    "schedule_form": True,
    "simple": True,
    "attacker_penalty_factor": 3,
    "defender_penalty_factor": 3
}
general_sum_kwargs = {
    "general_sum": True,
    "attacker_feature_value": 1,
    "defender_feature_value": 100,
    "defender_step_cost": 0,
    "alpha": 0.5
}

# Create and generate ISG instance
```

```
isg = InfraSecurityGame(df_combined, ny_blocks_gdf, INFRA_WEIGHTS,
    bbox=bbox_downtown_large)
isg.generate(
    num_attackers=1,
    num_defenders=3,
    home_base_assignments=[(fifth_precinct, booking_station, troop_nyc
        , first_precinct, police_plaza) for _ in range(3)],
    num_timesteps=8,
    defense_time_threshold=1,
    force_return=True,
    generate_utility_matrix=False,
    generate_actions=False,
    **schedule_form_kwargs,
    **general_sum_kwargs
)
```

## D.2 Sparsity Experiment Parameters

**General Setup.** All experiments were conducted on Green Security Game instances with the following fixed parameters unless otherwise specified:

- **Number of clusters**: 10
- **Grid dimensions**: $7 \times 7$
- **General-sum disabled**: All games were run as zero-sum games
- **Defense time threshold**: 1 for NFG, 2 for SFG
- **Attacker schedule form coverage scaler penalty factor (SFG only)**: 5
- **Defender schedule form coverage scaler penalty factor (SFG only)**: 5
- **Home base options**: Kabo Djembé (`2.0532, 16.0857`), Bomassa (`2.2037, 16.1871`), Inner Post (`2.2000, 15.9800`)

**NFG Sparsity Experiments.**

- **NFG (7 Timestep)**: 1 attackers, 1 defender, force return disabled, 9,075 defender actions.
- **NFG (8 Timestep)**: 1 attacker, 1 defender, force return disabled, 41,479 defender actions.

**SFG Sparsity Experiments.**

- **SFG (7 Timestep)**: 1 attacker, 2 defenders, force return enabled, 841 defender actions.
- **SFG (8 Timestep)**: 1 attacker, 2 defenders, force return enabled, 1,024 defender actions.
- **SFG (9 Timestep)**: 1 attacker, 2 defenders, force return enabled, 2,401 defender actions.

**Runtimes.** The total runtime for each sparsity experiment is calculated as the sum of the Nash equilibrium solver time and the cumulative runtime of all MIP-based defender best responses during the double oracle iterations. These values are reported in seconds:

- **NFG, 7 Timesteps**: Total runtime = 13.60s
  *(Nash: 1.03s, MIP: 12.56s)*
- **NFG, 8 Timesteps**: Total runtime = 741.13s
  *(Nash: 4.06s, MIP: 737.06s)*
- **SFG, 7 Timesteps**: Total runtime = 2.14s
  *(Nash: 0.10s, MIP: 2.04s)*
- **SFG, 8 Timesteps**: Total runtime = 2.74s
  *(Nash: 0.18s, MIP: 2.56s)*
- **SFG, 9 Timesteps**: Total runtime = 8.49s
  *(Nash: 0.29s, MIP: 8.21s)*

### D.3 Iterative Experiment Parameters

**Experiment Parameters.** All experiments were conducted on zero-sum Security Game instances using the following fixed parameters unless otherwise specified:

- **General-sum disabled**: All games were run as zero-sum games.
- **Attacker coverage penalty (SFG only)**: 5 (GSG), 3 (ISG).
- **Defender coverage penalty (SFG only)**: 5 (GSG), 3 (ISG).

**GSG NFG Experiment.**

- **Game Type**: Normal-form
- **Number of clusters**: 10
- **Grid dimensions**: $7 \times 7$
- **Number of attackers**: 1
- **Number of defenders**: 1
- **Timesteps**: 7
- **Defense time threshold**: 1
- **Force return**: Disabled
- **Home base options**: Kabo Djembé (2.0532, 16.0857), Bomassa (2.2037, 16.1871), Inner Post (2.2000, 15.9800)

**GSG SFG Experiment.**

- **Game Type**: Schedule-form (general schedules)
- **Number of clusters**: 10
- **Grid dimensions**: $7 \times 7$
- **Number of attackers**: 1
- **Number of defenders**: 2
- **Timesteps**: 7
- **Defense time threshold**: 1
- **Force return**: Enabled
- **Home base options**: Same as GSG NFG

**ISG NFG Experiment.**

- **Game Type**: Normal-form
- **Number of attackers**: 1
- **Number of defenders**: 1
- **Timesteps**: 7
- **Defense time threshold**: 1
- **Force return**: Enabled
- **Home base options**: First Precinct (40.7204, -74.0070), Fifth Precinct (40.7163, -73.9974), Police Plaza (40.7124, -74.0017), Troop NYC (40.7166, -74.0064), Booking Station (40.7162, -74.0010)

**ISG SFG Experiment.**

- **Game Type**: Schedule-form (general schedules)
- **Number of attackers**: 1
- **Number of defenders**: 2
- **Timesteps**: 7
- **Defense time threshold**: 1
- **Force return**: Enabled
- **Home base options**: Same as ISG NFG

**Regret Matching Algorithm Settings.** Each variant of regret matching was parameterized as follows:

- **RM** (vanilla regret matching): `runtime=120, interval=5, iterations=10000, averaging=0, alternations=False, plus=False, predictive=False`
- **RM+** (regret matching + with linear averaging and alternations): `runtime=120, interval=5, iterations=10000, averaging=1, alternations=True, plus=True, predictive=False`
- **PRM+** (predictive regret matching + with quadratic averaging and alternations): `runtime=120, interval=5, iterations=10000, averaging=2, alternations=True, plus=True, predictive=True`

### D.4  Stackelberg Experiment Parameters

**Stackelberg Setup.** All Stackelberg Security Game experiments were run as general-sum schedule-form games. The following common settings were used unless otherwise noted:

- **Schedule form**: Enabled
- **General-sum**: Enabled
- **Timesteps**: 7
- **Defense time threshold**: 1
- **Force return**: Enabled
- **Escape point logic**: Used to scale attacker values based on proximity (via `alpha` parameter)

**GSG SSE Experiments.** All GSG Stackelberg experiments used the same map: 10 centroid-scored clusters on a $7 \times 7$ grid, with the Sangha River (`(2.2837, 16.1628)` to `(2.053, 16.0662)`) as the escape line. Defenders began at the same set of three home bases used in all GSG experiments.

- **Simple Schedules**: Enabled
- **Number of attackers**: 1
- **Number of defenders**: 2
- **Attacker coverage scaler**: 5
- **Defender coverage scaler**: 5
- **Attacker target value**: 2350
- **Defender target value**: 22966
- **Defender step cost**: 0
- **Escape proximity scaling factor**: $\alpha = 1$

**General SSE GSG Setting:** Same as above, but:

- **Simple Schedules**: Disabled
- **Defender step cost**: 1.17 (realistic labor, fuel, and equipment cost per km)

**ISG SSE Experiments.**   All ISG Stackelberg experiments used the same Chinatown-area power grid graph with a fixed escape point at Brooklyn Bridge (40.7124, -74.0049). Defenders were allowed to start at any of the same 5 precinct locations used in all ISG experiments.

- **Simple Schedules**: Enabled

- **Number of attackers**: 1

- **Number of defenders**: 3

- **Attacker coverage scaler**: 3

- **Defender coverage scaler**: 3

- **Attacker target value**: 1

- **Defender target value**: 100

- **Escape proximity scaling factor**: $\alpha = 0.5$

**General SSE ISG Setting:** Same as above, but:

- **Defender step cost**: 1 (realistic patrol cost in USD per Manhattan block)

**Step Cost Estimation.**   We computed defender step costs based on realistic patrol expenses in each setting [46, 10, 14, 15, 34]:

- **GSG Step Cost**: Estimated at $1.17 per kilometer. This combines ranger labor ( $0.65/km, based on $345/month salaries, team size 5, and 15 km/h patrol speed) with equipment and supply costs (totaling  $0.29/km) and fuel costs ( $0.17/km). Resulting in roughly $1.17/km.

- **ISG Step Cost**: Estimated at $1.00 per Manhattan block. Based on NYPD officer pay ($33/hour), typical patrol speed (5 mph = 0.01 hr/block), and average patrol team size of 2, labor alone costs about $0.70/block. Adding estimated fuel ($0.02/block) and equipment costs ($0.15/block), and considering that some patrols have more than 2 officers, the total was roughly $1 per block.

**Elephant Values in USD.**   Defender elephant value of $22,966 reflects the one year eco-tourism value of an elephant. The attacker value is computed as an elephant's 2 tusks $\times$ 100 pound tusk average $\times$ $26/kg on the Cameroonian black-market is roughly $2,350 [41, 17]. These values serve to normalize the payoffs. Because they are constant scalers on target values, they do not tangibly impact the equilibria. However, the other general sum factors like path costs and escape proximities do have material impact on the equilibria as they decouple target values nonlinearly.

## D.5   Schedule Randomization

Schedules were randomized by assigning each defender the same number of schedules as in the real instance, with each schedule containing as many randomly selected targets equal to the ceil of the average real schedule length.

## D.6   Additional Experiment Results

Below are additional iterative and Stackelberg experiment results from our realistic security game instances, illustrating the strategic complexity and large relative support sizes characteristic of settings with enhanced realism through the GUARD framework. All home base, scaling factors, and general sum (for Stackelberg) parameter settings are equal to those used in the experiments in the main text and detailed in Appendices D.3 and D.4.

| Form | C | Grid | $\mathcal{T}$ | A | D | $\delta$ | FR | S | Time (s) | DO Size | Iters |
|------|---|------|------|---|---|---|----|---|----------|---------|-------|
| NFG | 12 | $10 \times 10$ | 10 | 1 | 3 | 1 | F | 12 | 98.25 | $45 \times 12$ | 46 |
| NFG | 12 | $9 \times 9$ | 10 | 3 | 3 | 1 | F | 12 | 40.79 | $37 \times 41$ | 41 |
| NFG | 12 | $8 \times 8$ | 10 | 2 | 3 | 1 | T | 11 | 72.71 | $32 \times 31$ | 33 |
| NFG | 11 | $8 \times 8$ | 9 | 2 | 3 | 2 | T | 11 | 32.93 | $27 \times 27$ | 28 |
| NFG | 10 | $7 \times 7$ | 8 | 2 | 3 | 2 | F | 10 | 23.07 | $28 \times 27$ | 29 |
| SFG | 12 | $8 \times 8$ | 9 | 1 | 2 | 2 | T | 9 | 0.11 | $31 \times 10$ | 35 |
| SFG | 10 | $10 \times 10$ | 7 | 1 | 3 | 2 | T | 9 | 0.17 | $23 \times 9$ | 24 |

Table 5: **Results from GSG iterative double oracle experiments**. Columns: C = Clusters, $\mathcal{T}$ = Timesteps, A = Attackers, D = Defenders, $\delta$ = Defense Time Threshold, FR = Force Return, S = Double Oracle Support, DO Size = Final action space dimensions of the DO subgame ($|A_d| \times |A_a|$), Iters = Iterations to Converge.

| Form | $\mathcal{T}$ | A | D | $\delta$ | FR | S | Time (s) | DO Size | Iters |
|------|------|---|---|---|----|---|----------|---------|-------|
| NFG | 9 | 3 | 3 | 2 | F | 16 | 671.05 | $27 \times 33$ | 33 |
| NFG | 10 | 2 | 3 | 2 | F | 15 | 1661.28 | $31 \times 32$ | 32 |
| NFG | 9 | 3 | 3 | 1 | F | 15 | 2303.54 | $41 \times 43$ | 43 |
| NFG | 10 | 1 | 3 | 2 | F | 14 | 1494.30 | $47 \times 17$ | 47 |
| NFG | 8 | 2 | 3 | 1 | F | 14 | 162.24 | $27 \times 28$ | 28 |
| SFG | 9 | 3 | 2 | 1 | T | 12 | 0.28 | $37 \times 13$ | 36 |
| SFG | 6 | 3 | 1 | 1 | T | 10 | 0.16 | $28 \times 12$ | 28 |

Table 6: **Results from ISG iterative double oracle experiments.**

| Form | C | Grid | $\mathcal{T}$ | A | D | $\delta$ | FR | S | Time (s) | Def. Utility |
|------|---|------|------|---|---|---|----|---|----------|--------------|
| SFG $\rightarrow$ NFG | 10 | $7 \times 7$ | 9 | 1 | 3 | 2 | T | 9 | 23.62 | -0.133 |
| SFG $\rightarrow$ NFG | 10 | $8 \times 8$ | 8 | 1 | 3 | 1 | T | 9 | 5.08 | -0.150 |
| SFG $\rightarrow$ NFG | 10 | $10 \times 10$ | 7 | 1 | 2 | 1 | T | 9 | 0.19 | -0.230 |
| SFG $\rightarrow$ NFG | 10 | $7 \times 7$ | 9 | 1 | 1 | 1 | T | 8 | 0.03 | -0.285 |
| SFG $\rightarrow$ NFG | 8 | $8 \times 8$ | 9 | 1 | 2 | 2 | T | 8 | 0.06 | -0.204 |

Table 7: **SSE experiment results for GSGs (general schedules, general sum, multiple LP SSE solver).** SFG $\rightarrow$ NFG indicates normal form matrix expansion. Columns: C = Clusters, $\mathcal{T}$ = Timesteps, A = Attackers, D = Defenders, $\delta$ = Defense Time Threshold, FR = Force Return, S = Support Size, Def. Utility = Defender max utility at equilibrium.

| Form | $\mathcal{T}$ | A | D | $\delta$ | FR | S | Time (s) | Def. Utility |
|------|------|---|---|---|----|---|----------|--------------|
| SFG $\rightarrow$ NFG | 7 | 1 | 3 | 1 | T | 14 | 838.88 | -0.408 |
| SFG $\rightarrow$ NFG | 9 | 1 | 3 | 2 | T | 13 | 762.70 | -0.408 |
| SFG $\rightarrow$ NFG | 8 | 1 | 3 | 1 | T | 13 | 2856.51 | -0.404 |
| SFG $\rightarrow$ NFG | 7 | 1 | 3 | 2 | T | 12 | 113.75 | -0.422 |
| SFG $\rightarrow$ NFG | 6 | 1 | 3 | 1 | T | 10 | 224.72 | -0.421 |

Table 8: **SSE experiment results for ISGs (general schedules, general sum, multiple LP SSE solver).**

## D.7   Infra Weights (for ISGs)

Listing 5: Infrastructure Security Game feature base weights, manually assigned based on qualitative assessments of each feature's relative criticality. While the default values serve as reasonable approximations, the framework is designed to allow users to adjust the weights they use to reflect application/domain-specific valuations.

```python
INFRA_WEIGHTS = {
    # Power Infrastructure
    "plant": 1.5,
    "generator": 1.35,
    "solar_generator": 0.95,
    "substation": 1.45,
    "transformer": 1.25,
    "tower": 1.1,
    "pole": 0.85,
    "line": 1.0,
    "minor_line": 0.9,
    "cable": 0.95,
    "switchgear": 1.2,
    "busbar": 0.8,
    "bay": 0.85,
    "converter": 1.05,
    "insulator": 0.75,
    "portal": 0.75,
    "connection": 0.7,
    "compensator": 1.0,
    "rectifier": 0.95,
    "inverter": 0.95,
    "storage": 0.9,

    # Healthcare
    "hospital": 1.5,
    "clinic": 1.35,

    # Education
    "school": 1.25,
    "university": 1.4,

    # Water & Sanitation
    "water_works": 1.45,
    "wastewater_plant": 1.4,

    # Government & Emergency Services
    "fire_station": 1.3,
    "police": 1.4,
    "courthouse": 1.2,

    # Critical Infrastructure
    "bunker_silo": 1.0,

    # Communications
    "communications_tower": 1.25,
}
```

