# OpenReview forum: "GUARD: Constructing Realistic Two-Player Matrix and Security Games for Benchmarking Game-Theoretic Algorithms"
_NeurIPS.cc/2025/Datasets_and_Benchmarks_Track — NeurIPS 2025 Datasets and Benchmarks Track spotlight_

### Official Review · Reviewer_ATEE · 2025-06-07

**Rating:** 5
**Confidence:** 3

**Summary:**

The paper points out that security games are difficult to use for evaluating game-theoretic algorithms due to the lack of target value data, and proposes the GUARD framework to generate realistic matrix and security game instances using open-source data (such as animal movement and infrastructure data). It theoretically analyzes the sparsity defects of equilibria in random games, and experiments show that the generated games can more effectively evaluate algorithms than random baselines, providing practical benchmarks for game-theoretic algorithms.

**Dataset Code Accessibility:**

Partly

**Ethical Considerations:**

No, there are no or only very minor ethics concerns

**Limitations Weaknesses:**

I consider this to be a solid piece of work that addresses important issues and makes meaningful progress. The limitations acknowledged in the paper—such as scalability—do not significantly detract from its contributions. However, the framework still has substantial room for improvement in terms of usability.

Currently, the process of importing new data and maps lacks standardized interfaces, requiring users to manually process raw data formats and make extensive modifications to the source code for experimental configuration. This presents a steep learning curve, particularly for users outside the immediate research domain. Furthermore, the framework does not offer structured API documentation. The definitions of core functions—such as schedule generation algorithms and utility matrix construction—are scattered across code snippets in the appendix, making it difficult for users to quickly integrate new solution strategies, such as custom LLM-based agents or reinforcement learning methods.

To enhance the framework’s usability and broaden its impact, I recommend prioritizing the following improvements:

1. Providing one-click tools for data import and preprocessing;

2. Releasing modular API documentation to clearly specify the input-output behaviors of key components, particularly the Game and Solver classes.

**Strengths Contributions:**

The paper boasts remarkable strengths. The proposed GUARD framework innovatively leverages open-access data (such as animal movement trajectories, demographic statistics, and infrastructure data) to generate high-fidelity security game instances, filling the void of missing real-world scenario data. Theoretically, it systematically uncovers for the first time the sparsity and degeneracy flaws in equilibrium solving of random games, providing a theoretical basis for benchmark design. Experiments validate the framework's effectiveness by comparing the performance of multiple algorithms in real versus random games.

---

> ### Author Rebuttal · Authors · 2025-07-30
>
> Thank you very much for your thoughtful and constructive review. We greatly appreciate the time and care you took to engage with our work. We would like to respond to the usability-related concerns you raised, specifically regarding data accessibility and documentation.
>
> As you noted, the GUARD framework uses a range of publicly available datasets, including Movebank animal movement data, U.S. Census statistics, and OpenStreetMap infrastructure data. We agree that improving the ease of integrating such data would significantly enhance the framework’s usability. However, each data source presents its own access challenges. For example, Movebank requires either navigating a world map GUI or published study indexes, Census data access depends on jurisdiction and year, and OpenStreetMap queries require structured formats, executable through services like Overpass Turbo. Even if as a result of these complexities the single click pipeline for automated data access is nontrivial, we pledge to streamline the process by adding detailed step-by-step instructions for accessing and preprocessing data from each source. These instructions will be included in the README of the public GUARD repository.
>
> To further ease use, the framework also includes a suite of pre-generated realistic game instances, which we expect will cover the majority of evaluation needs for researchers interested in testing new algorithms. These out of the box games will eliminate the need for raw data handling in many use cases. We also plan to develop a user-friendly GUI or command-line interface that simplifies custom game generation. Users will be able to configure high level parameters via simple inputs, while the back end manages everything data-related.
>
> Regarding documentation, we agree that structured API-level guidance would be a useful addition. While the code currently includes Google-style docstrings for all major functions, we plan to add full documentation outlining the behavior, inputs, and outputs of key components, specifically for core elements such as the Game and Solver classes which you bring up. This documentation will be organized into dedicated tabs in the public repository and supplemented with thorough usage examples.
>
> Once again, thank you for your helpful and encouraging feedback. Please let us know if you have any further questions or concerns.

---

> > ### Comment · Reviewer_ATEE · 2025-08-01
> >
> > Thank you for your detailed response. We hope you can complete the promised content revisions as soon as possible. Based on the current situation, I am inclined to maintain my original rating.

---

### Official Review · Reviewer_owD1 · 2025-07-01

**Rating:** 5
**Confidence:** 3

**Summary:**

The paper introduces GUARD, a framework for generating realistic zero-sum two-player matrix and security games using publicly available data, addressing the limitations of synthetic benchmarks. Traditional benchmarks rely on randomly generated utilities and lack real-world grounding. In contrast, GUARD leverages data such as animal movement, infrastructure locations, and population statistics to construct meaningful game structures and utility functions. These data sources define not only the underlying graph and targets but also the utilities for both attackers and defenders. GUARD supports multiple game formats and solvers, enabling more accurate benchmarking of game-theoretic algorithms in realistic security scenarios.

**Dataset Code Accessibility:**

Yes

**Dataset Code Comments:**

The dataset and source code have been open-sourced on GitHub.

**Ethical Considerations:**

No, there are no or only very minor ethics concerns

**Final Justification:**

The author has addressed my questions, so that I will maintain my evaluation.

**Limitations Weaknesses:**

1. While the paper critiques traditional benchmarks for relying on purely random utilities, it does not sufficiently discuss GAMUT, a widely used game generator. GAMUT produces games with predefined structures (e.g., zero-sum, coordination), and although utilities within those structures are randomized, they are not entirely unstructured. A more detailed comparison explaining why and how GUARD offers advantages over GAMUT would strengthen the paper.
2. GUARD currently supports only two-player games. However, many real-world security scenarios involve teams of attackers and defenders. It would be valuable to discuss the potential for extending the framework to support team-based (multi-agent) games.
3. In the experimental section, the paper emphasizes the sparsity of strategies in generated games but does not clearly explain why this property matters. It would improve clarity to explicitly state the significance of sparsity.

**Strengths Contributions:**

1. The paper leverages publicly available, real-world datasets—such as animal movement and infrastructure data—to generate security games that reflect actual environments, addressing the limitations of traditional benchmarks that rely on randomly generated or purely synthetic utilities.
2. The framework utilizes public data not only to construct the game graph and target locations but also to assign realistic utility values to both attackers and defenders,.
3. GUARD supports both normal-form and Stackelberg formulations.

---

> ### Author Rebuttal · Authors · 2025-07-30
>
> Thank you very much for your thoughtful and constructive review. We appreciate the time you took to engage with our work and would like to address the points you raised regarding differentiation from GAMUT, support for multiplayer games, and the discussion of strategy sparsity.
>
> Regarding GAMUT, another framework alternative that focuses on different classes of games, you noted that our paper could benefit from a more in depth explanation of the distinctions between the two frameworks. While we do not consider GAMUT a direct competitor due differences in objective and methodology, we agree that a more explicit comparison would be helpful for readers. While GAMUT does generate games with structured types (e.g., coordination, zero-sum), its utility assignments remain randomized and detached from any real-world dynamics. In contrast, GUARD uses open-access data sources to define not only the game structure but also the utility values, enabling a more realistically-grounded game environment. Additionally, the types of games offered by GAMUT differ substantially from those in GUARD, with GAMUT’s graph games lacking the capacity to incorporate real-world data into target values or network structure. We will add a detailed comparison with GAMUT in the appendix to clarify these distinctions in the two frameworks.
>
> On the point about two-player limitations, we agree that extending GUARD to handle multiplayer or team-based scenarios is a valuable direction. In fact, since the initial submission, we have implemented and released a basic form of multiplayer support within the GUARD framework. This functionality enables multiple attacker and defender units with distinct incentives, allowing for richer strategy spaces and interactions. While still in an early stage, we intend to continue expanding this capability to support more complex multi-agent game formats in future versions.
>
> Finally, we appreciate your comment on the discussion of sparsity. While we highlight empirical observations regarding strategy sparsity in games generated by GUARD, the motivation behind its significance could be more clearly explained. In particular, we view sparsity as an indicator of how realistically a benchmark game reflects strategic complexity. Extremely sparse strategies may suggest degeneracy or oversimplification in the game environment, which is often a limitation of synthetic benchmarks, specifically for general sum games. Conversely, overly dense strategies may be impractical in real-world applications. GUARD aims to generate games that produce strategy profiles with realistic levels of complexity. We pledge to revise the paper to clarify these concepts and their relevance for evaluating algorithmic performance.
>
> Thank you again for your insightful feedback. Please don’t hesitate to reach out with any further questions or recommendations.

---

> > ### Comment · Reviewer_owD1 · 2025-08-04
> >
> > Thank you for your response. It addressed my questions.

---

### Official Review · Reviewer_R4vd · 2025-07-02

**Rating:** 4
**Confidence:** 3

**Summary:**

This work introduces GUARD an open-source framework for generating realistic matrix and security game instances based on open-access data. Motivated by the absence of practical security games in standard game-theoretic benchmarks, GUARD fills a critical gap by enabling reproducible evaluation of equilibrium-finding algorithms in real-world-inspired settings. Drawing on sources such as animal movement data and infrastructure demographics, GUARD supports both custom and pre-defined game instances, and exports in formats compatible with OpenSpiel and Gambit. The paper also presents a theoretical and empirical critique of using randomly generated games as benchmarks, showing that Stackelberg equilibria in such games tend to be sparse and degenerate unlike those in realistic scenarios.

**Additional Feedback:**

Try to address the limitations mentioned above.

**Dataset Code Accessibility:**

Yes

**Ethical Considerations:**

No, there are no or only very minor ethics concerns

**Final Justification:**

I have read all the reviews, the authors’ rebuttal, and the final remarks. I believe this is a strong paper, and the authors have adequately addressed my concerns. I would like to retain my original score.

**Limitations Weaknesses:**

Authors explicitly provide the limitations of benchmarking on random games. Apart from these feel there are some limitations of this work:

1. Translating real-world features (e.g., animal presence, infrastructure value) into target utilities involves subjective or heuristic design choices. These mappings may not capture all the nuanced strategic considerations present in actual security deployments.

2. While GUARD instances are more realistic than random games, they are still abstractions. Algorithms that perform well on GUARD benchmarks may not directly generalize to all operational environments without further validation.

3. Although GUARD addresses security and matrix games, it may not cover all important classes of strategic interactions such as extensive-form games with incomplete information or multi-stage signaling games.

**Strengths Contributions:**

1. Authors claim that Gaurd is flexible, open-source tool for generating realistic matrix and security games from public data sources.

2. They have a repository of ready-made, security-inspired game instances to support reproducible research.

3. It also Integrates well with popular tools  like OpenSpiel, Gambit, GraphChase.

4. The main contribution is its theoretical analysis which proves that a random Stackelberg games yield degenerate equilibria that favor the defender unrealistically.

4. Authors demonstrate a significant differences in equilibrium structure between random and realistic games, across various game settings.

---

> ### Author Rebuttal · Authors · 2025-07-30
>
> Thank you for your thoughtful and constructive review. We appreciate the time you took to read and engage with our work. Below, we address your comments regarding the use of heuristics in modeling game mechanics, the abstract nature of GUARD instances, and the scope of game types currently supported.
>
> You note that translating real-world features into target utilities necessarily involves heuristic design choices that may not fully capture the complexity of actual strategic deployments. This is a valid point, and we acknowledge that our current approach relies on parameter mappings that serve as informed approximations of real-world incentives. However, we emphasize that these parameters were grounded in empirical data and literature, including: black-market values of ivory, the estimated economic value of an elephant to tourism, ranger patrol costs, animal population densities, apprehension rates, and infrastructure proximity to population centers. Our objective was to approximate domain-relevant incentives as closely as possible, and in doing so, we prioritized transparency and flexibility.
>
> Importantly, all parameters in GUARD are user-configurable. The framework is designed to allow researchers to inject domain expertise when defining utilities and game structures. We agree that further reducing the friction of this customization process would be valuable, and we are currently exploring options for a more user-friendly interface or configuration toolkit to support more streamlined tuning of game parameters.
>
> Regarding your point that GUARD games remain abstractions, we fully agree. No benchmark, however data-driven, can perfectly replicate the complexity of real-world environments. However, we believe GUARD represents a substantial improvement over commonly used alternatives. While GUARD games are not direct substitutes for deployment-specific simulations, they offer a more meaningful proxy for many research use cases and improve upon the synthetic benchmarks currently dominating the literature.
>
> Finally, you raise the point that GUARD currently focuses on matrix and Stackelberg security games, and does not yet support other important classes such as extensive-form or signaling games. This is a deliberate choice: we chose to focus initially on security games due to their real-world relevance and widespread use in algorithmic game theory, and the fact that the existence of realism-based benchmarks for such a common application-based research area are notably absent. That said, we are actively considering extensions to broader classes of games, including multi-stage or incomplete-information formats, and we view this as a natural direction for future work.
>
> Thank you again for your constructive feedback. Please feel free to reach out with any additional questions or suggestions.

---

### Official Review · Reviewer_kmZ9 · 2025-07-20

**Rating:** 4
**Confidence:** 4

**Summary:**

This submission presents GUARD, a framework for constructing two-player matrix and security games using real-world datasets for algorithm benchmarking. The work contributes a data-driven framework using existing open data, theoretical analysis showing limitations of random game benchmarking, empirical evaluation comparing realistic versus random games across standard equilibrium algorithms, and a library of pre-configured game instances based on authentic scenarios.

**Dataset Code Accessibility:**

Yes

**Ethical Considerations:**

No, there are no or only very minor ethics concerns

**Final Justification:**

Most of my concerns are solved.

**Limitations Weaknesses:**

The paper's introduction motivates the work with high-impact applications such as security at Los Angeles International Airport, yet the experimental validation relies on simple scenarios without any difference from other papers. This disconnect between ambitious motivating examples and limited experimental scope undermines the work's persuasive impact.

The research represents an applied engineering contribution that constructs a framework for generating game instances by applying established game-theoretic concepts (Nash, Stackelberg) to real-world datasets from sources including Movebank and the U.S. Census. Without novel theoretical contributions, this work constitutes an incremental advance rather than a fundamental breakthrough as claimed, falling short of the acceptance threshold.

**Strengths Contributions:**

GUARD is an open-source framework that generates realistic security game benchmarks from real-world, open-access data, addressing the well-known limitations of  random games typically used for evaluating game-theoretic algorithms. The framework demonstrates that realistic game instances produced through GUARD elicit significantly more complex and strategically diverse algorithmic behavior compared to traditional random baselines.

---

> ### Author Rebuttal · Authors · 2025-07-30
>
> Thank you for your detailed review and for taking the time to engage with our submission. We appreciate the opportunity to clarify the intent and contributions of our work, particularly regarding the significance of our game scenarios and the novelty of our theoretical results.
>
> You raise a concern about a perceived mismatch between the high-impact applications discussed in the introduction (e.g., airport security) and the experimental validation. While we do reference airport security games to contextualize the broader importance of security game research, our experimental focus is deliberately centered on green security games and infrastructure security games. We chose to emphasize these settings for several reasons:
>
> - $\textbf{Green security games (GSGs)}$ represent one of the most well-developed and empirically validated applications of Stackelberg security game models in both academic literature and real-world deployment. They have a rich history of implementation, including the PAWS system (Protection Assistant for Wildlife Security), and have been the subject of foundational work by researchers such as F. Fang and M. Tambe. The GUARD framework builds directly on this tradition, using real ecological data to instantiate GSGs with greater fidelity than previously possible.
>
>
> - $\textbf{Urban Infrastructure Security}$ is a natural extension of the Stackelberg model and allows for applications across a wide range of public safety and facility protection scenarios. Additionally, the infrastructure security game model in the GUARD framework (as well as the GSG model) is capable of building generalized security games of realistic real-world size and complexity when sufficient data is available. In our experiments, we rely on publicly available data (e.g., from OpenStreetMap and the U.S. Census) to construct these games. Unfortunately, comparable high-quality data for modeling realistic airport security scenarios is not readily available in the public domain, motivating our initial focus on modeling GSGs. We do, however, plan to support such applications in future iterations of GUARD, and users with access to private airport-relevant datasets can already leverage our framework to construct those games.
>
>
> We respectfully disagree with the claim that our work lacks novelty and theoretical results. Our work differs from prior research in two essential ways:
>
> - $\textbf{First}$, unlike earlier efforts, our goal is not to propose new solution algorithms for security games but to fundamentally improve the benchmarking infrastructure through the introduction of realistic, reproducible, and data-driven game instances. To the best of our knowledge, no existing framework offers comparable support for generating security or matrix games directly from real-world open-access data.
>
> - $\textbf{Second}$, our paper does present original theoretical contributions. Specifically, in Section 3, we prove that Stackelberg equilibria in randomly generated games, both in matrix form and in security game settings, exhibit a bias toward degenerate or overly sparse strategy profiles that overestimate defender performance. Theorems 1 and 2 on page 3 formalize this finding and are accompanied by full proofs in the appendix on pages 13 to 22. These results, which to our knowledge have not appeared elsewhere in the literature, offer a novel critique of a core assumption underlying decades of equilibrium algorithm evaluation using synthetic benchmarks. We respectfully believe this constitutes a meaningful theoretical contribution that advances the understanding of strategic behavior in randomized environments.
>
>
> In summary, while our experimental domains draw on established security game applications, we introduce both a novel framework and original theoretical insights that address well-known limitations in current benchmarking practices. We appreciate your feedback and welcome further questions or clarifications.

---

> > ### Comment · Reviewer_kmZ9 · 2025-08-05
> >
> > Although I think some points are overclaimed in this paper, I will increase my score because most of the concerns are solved and the efforts of the authors.

---

### Decision · Program_Chairs · 2025-09-18

**Decision:**

Accept (spotlight)

**Comment:**

This paper received very consistent and positive reviews both before and after the rebuttal (slight increase after the rebuttal).

The paper presents GUARD, an open-source framework that generates realistic security game benchmarks from real-world, open-access data; it shows the limitations of randomly generated games typically used in game-theoretic studies/papers, and that GUARD generates game instances that elicit significantly more complex and strategically diverse algorithmic behavior compared to traditional random baselines. Thus the paper produces a useful benchmark -- this is the main reason for acceptance.

Limitations include the fact that GUARD as a game instance generator also relies on assumptions (e.g., the agent's utility) some of which can be very subjective.

===== FINAL UPDATE FROM DB Track PCs ====

The final decision for this paper has been taken by the program chairs after consultation with the SACs. All Senior Area Chairs have ranked papers according to the feedback from the AC during the review process. We decided to leave the original meta-review to reflect the opinion of the AC in light of the initial discussions with reviewers and SAC.